# Revisiting Viral RNA-Dependent RNA Polymerases: Insights from Recent Structural Studies

**DOI:** 10.3390/v14102200

**Published:** 2022-10-06

**Authors:** Kavitha Ramaswamy, Mariya Rashid, Selvarajan Ramasamy, Tamilselvan Jayavelu, Sangita Venkataraman

**Affiliations:** 1Department of Biotechnology, Anna University, Sardar Patel Road, Guindy, Chennai 600025, India; kavirams23@gmail.com (K.R.); tmlsln.j@gmail.com (T.J.); 2Taiwan International Graduate Program, Molecular Cell Biology (National Defense Medical Center and Academia Sinica), Taipei 115, Taiwan; mariyarashid96@gmail.com; 3National Research Center for Banana, Somarasempettai−Thogaimalai Rd, Podavur, Tamil Nadu 639103, India; selvarajanr@gmail.com

**Keywords:** replication, polymerase, cofactors, structure, RNA viruses, inhibitors

## Abstract

RNA-dependent RNA polymerases (RdRPs) represent a distinctive yet versatile class of nucleic acid polymerases encoded by RNA viruses for the replication and transcription of their genome. The structure of the RdRP is comparable to that of a cupped right hand consisting of fingers, palm, and thumb subdomains. Despite the presence of a common structural core, the RdRPs differ significantly in the mechanistic details of RNA binding and polymerization. The present review aims at exploring these incongruities in light of recent structural studies of RdRP complexes with diverse cofactors, RNA moieties, analogs, and inhibitors.

## 1. Introduction

RdRPs (EC 2.7.7.48.) are ancient enzymes that catalyze the formation of phosphodiester bonds between ribonucleotides in an RNA template-dependent fashion using divalent metal ions. The reaction involves the nucleophilic attack of the 3’-hydroxyl group of the nascent RNA on the α-phosphate of the incoming nucleoside triphosphate (NTP), resulting in the formation of a new phosphodiester bond and the release of a pyrophosphate moiety (Figure 1A). Together with other viral and host proteins, the RdRPs participate in RNA synthesis, discriminating replication and transcription events, and incorporating the 5’ cap and the 3’ poly A tail [1]. RdRP is the sole enzyme that exhibits significant similarity to ancestral t-RNA sequences, suggesting a descendance from them [2]. Most of the viral RdRPs were identified based on comparative sequence analysis and are highly conserved in the viral genome [3]. The RdRPs have strikingly similar domain and motif arrangements, demonstrating how evolutionary conservation manifests itself at the level of structure [4].

RdRPs are classified as alpha and beta proteins and fold majorly into three subdomains, namely the fingers, thumb, and palm (Figure 1B) [3,5,6,7,8]. The majority of the structurally conserved components are located in the palm subdomain, which is sandwiched between the fingers and the thumb subdomains. The palm harbors the catalytic aspartates and the three-stranded RNA-recognizing motif (RRM). The variable thumb subdomain interacts with the primer while the fingers play a crucial role in template binding [1,3]. The RdRPs exhibit “close-hand” conformation as opposed to DNA polymerases owing to the projecting loops (“fingertips”) that make the fingers and the thumb subdomains appear interconnected [7,9]. The structural features of the RdRP result in the formation of two channels that converge at the active site (Figure 1B). The RNA template gains access through the primary channel while the incoming NTPs enter through the secondary channel. To further their ability to service their needs, several RdRPs possess other channels, including the RNA exit channel [6,7,10,11]. Seven conserved motifs, A through G, were discovered by comparing the structures and sequences of RdRPs that are implicated in the binding of metal ions, NTPs, and RNA (Figure 1B) [12,13]. Motifs A through E reside in the fingers subdomain, while motifs G and F are nested in the palm subdomain. For the group III and IV viruses, a further motif H was found in the thumb subdomain. Motif A houses the first aspartate of the catalytic motif DX2-4D. The motif B of the palm subdomain aids in the binding of the template RNA and in substrate discrimination. The conserved GDD motif, which is used to bind metal ions, is located in the loop of motif C. Due to the presence of a conserved glycine, motif D functions as an anchor for conformational changes related to correct NTP binding. Motif E, commonly known as “the primer grip”, aids in the proper positioning of the primer’s 3′ hydroxyl group during catalysis. Motif F interacts with the phosphate group of the incoming NTP while the loop of motif G forms a part of the template entrance channel in group IV viruses [4,6,9,14]. 

The sizes of viral RdRPs vary greatly due to the presence of auxiliary domains that aid in catalysis. Those from the family *Reoviridae* and phage φ6 are 148 and 76 kDa, respectively, and possess extended N- and C-terminal domains in addition to the core [10,15]. The *picornaviral* 3D^pol^ is only ∼50 kDa [9,16], while the *flaviviral* NS5b is 100 kDa in size as it includes a fused methyltransferase (MTase) domain [11,17]. The *nidoviral* NiRAN (nidovirus RdRP-associated nucleotidyl transferase) domain spans over 350 residues and is linked to the C-terminus of the RdRP [18]. The RdRPs of the non-segmented single-stranded (ss) (-) RNA viruses, such as the *Vesicular stomatitis virus* (VSV) (240 kDa), have polyribonucleotidyl transferase (PRNTase) and MTase domains along with a ring-like core [19]. The segmented ss (-) RNA influenza virus replicase complex is ~260 kDa, comprising the polymerase acidic (PA), polymerase basic (PB), and RdRP subunits [20]. The structural conservation and variations in the RdRPs both within and across families, have been extensively analyzed [3,4,9,18,21,22]. In this article, we have reviewed the recent structural studies of RdRP complexes from different viral families in terms of modes of replication and inhibition.

## 2. The Business of Copying

The structural snapshots of RdRPs in association with nucleic acid moieties and analogs have greatly aided the understanding of their modus operandi. The exact geometry of the active site and the disposition of the template/product, different domains, motifs, and cofactors during the pre- and post-stages of catalysis are now better understood. The following subsections summarize these findings (May 2018 to present) (Table 1).

### 2.1. Cooperative Cofactors

A repertoire of self and co-opted host-encoded proteins assist the RdRP during genomic replication and transcription. These include the self-encoded helicases; capping enzymes; endonucleases; VPg; phosphoproteins; and the subverted proteins of the host, such as the translation factors, protein chaperones, RNA-modifying enzymes, and cellular proteins. Altogether, the assembly of the replication transcription complex (RTC) is a rather complex process driven by many factors for optimal replication in the infected cells.

In families such as *Potyviridae*, *Picornaviridae*, and *Caliciviridae,* the viral protein genome-linked (VPg) is covalently linked at the 5′ end of the genome and functions as a primer during replication [23,24,25,26]. The co-crystal structure of *Murine norovirus* (MNV) RdRP with VPg highlighted the role of the latter in both priming and multimerization during replication (PDB 5Y3D) [27]. In the presence of RNA, the MNV RdRP–VPg complex (1–73) clustered and displayed a tightly packed, ball-like configuration reinforced by the interactions of the RdRP with the disordered C-terminus of the VPg (Figure 2). The resulting ensemble not only offered a sizable interior area to trap ss RNA but also improved cooperative RNA binding for effective replication [27]. The structural investigation of the influenza polymerases from the human A/NT/60/1968 (H3N2) and avian A/duck/Fujian/01/2002 (H5N1) viruses also highlights the necessity of dimerization of the polymerase during genome replication (PDB 6QNW, 6QPF, 6QPG, 6QWL, 6QX3, 6QX8, 6QXE, 6RR7) [28].

Diverse cofactors are required by the coronaviral polymerases for effective transcription and replication of the large and complex coronaviral genomes. Hence, coronaviruses possess a highly processive polymerase (RdRP core with nsp8/nsp7), a system for proofreading (nsp14 with the 3’–5’ exoribonuclease activity), a helicase (nsp13) for unwinding the template–product duplex, and the MTases for capping (nsp14, nsp16) [29]. The structure of the replication transcription complex (RTC) of SARS-CoV-2 features two copies of the nsp8 flanking the nsp12 (PDB 7BW4, 6YYT, 7CYQ, 7C2K, 7BZF, 7BTF, 6NUR, 6NUS) [30,31,32,33,34]. The N-terminal helical extensions of the nsp8 act as “positively charged sliding poles”, facilitating the process of elongation and engaging with the exiting RNA duplex (Figure 3A) [30,31,33]. During replication, the two nsp8 copies show varying patterns of association with the nsp12 and nsp7. The pre- and the post-translocated states of the RTC reveal a 45° difference in the orientation of the N-terminal helical extensions of the two nsp8 copies (PDB 7C2K, 7BZF) [32]. The nsp7 and nsp8 also act as a “non-canonical primase”, mirroring the SARS-CoV-1 hexadecameric primase ring complex in the mode of RNA binding [32,35]. During the early stages of replication, the RTC binds to the double-stranded (ds) RNA helix in a sequence-independent manner via the 2’OH of the ribose, demonstrating the basis for the RNA selectivity (Figure 3A) (PDB 7BV1, 7BV2) [36]. Earlier studies of the RTC with nsp13 proposed the role of the latter in backtracking, thereby facilitating template switching, proofreading, or both during subgenomic RNA transcription (PDB 6XEZ, 7KRN, 7KRO, 7KRP) [37,38]. The association of RTC with two nsp13 moieties aided in transitioning between backtracking and accelerated RNA synthesis in response to cues at the RdRP’s active site via an allosterically controlled mechanism (PDB 7RDX, 7RDY, 7RDZ, 7RE0, 7RE1, 7RE2, 7RE3). The nsp13 achieved this extraordinary feat by means of four different conformational states [39]. Recently, Park et al. proposed a carefully orchestrated relay involving the NiRAN domain, nsp9, nsp13, and the MTases to cap the 5′ end of the genome (Figure 3B) (PDB 7THM) [40]. In the first step, the NiRAN domain transfers the 5′-product RNA to the amino terminus of nsp9. The RNAylated nsp9 is then attacked by the GDP, which is produced by the nsp13 via GTP hydrolysis, releasing the capped RNA and regenerating the unmodified nsp9. Finally, nsp14 and nsp16 generate a fully functional cap via RNA methylation. The STAT2 protein, a key player in inducing IFN responses during viral infections, was found to interact in a two-way manner with the nsp5 of the Dengue (DENV) and Zika viruses (Figure 3C) (PDB 6UX2, 6WCZ) [41]. The RdRP is associated with the NTD of STAT2 while its coiled-coil domain is nestled in a conserved inter-domain crevice between the methyltransferase and the RdRP domains. As both nsp5 and the interferon regulatory factor 9 compete for the same interaction surface of STAT2, the binding of the former resulted in the suppression of the IFN-induced expression of antiviral genes [41].

In the order *Mononegavirales*, the nucleocapsid-associated template is copied by the L protein with the aid of its cofactor, the tetrameric phosphoprotein (P), during genome replication and transcription [42,43]. The RdRP, cap-specific MTases, and GDP PRNTase domains of the ring-shaped L protein make it multifunctional. The RdRP domain of *Human metapneumovirus* (HMPV) comprises the canonical fingers–palm–thumb subdomains [44,45]. The L–P complexes were structurally characterized for the segmented HMPV (PDB 6U5O) [44], the *Respiratory syncytial virus* (RSV) (PDB 6PZK) [46], and the non-segmented *Parainfluenza virus 5* (PIV5) (PDB 6V85, 6V86, 6VAG) [47]. In the strikingly similar complexes, the helices and their extensions from the tetramerization domain of the P interact with the L and clasp it in a tentacle-like manner (Figure 4a). The P monomers associate with the L and the nucleocapsid through overlapping regions exhibiting distinct conformations and interactions [19,44,46]. The L–P complex structure of PIV5 revealed that the central oligomerization domain and the C-terminal X domain of the P attach to two distinct binding surfaces on the L protein. The active site of the MTase was positioned directly above the PRNTase domain, thereby reducing the distance between the former and the conserved HR motif of the latter by ~27 Å, representing the complex in a transcription-competent form [47]. The unique conformations of the MTase and the CTD relative to the RdRP domain in PIV5 might have implications for switching between genome replication and transcription (Figure 4B). The cryo-EM structures of the *Influenza C virus* (FluC) polymerase in complex with human and chicken acidic nuclear phosphoprotein 32 (ANP32) revealed that the protein served as a replication platform for the viral RNA (vRNA) (Figure 4C) (PDB 6XZD, 6XZG, 6XZP, 6XZQ, 6XZR, 6Y0C). In both complexes, two polymerase molecules were bridged by the amino-terminal leucine-rich repeat domain of ANP32 [48]. 

The *arenaviral* zinc-binding regulatory matrix protein (Z) synchronizes the assembly and budding of progeny viruses, negatively regulates RNA synthesis, and blocks host immunity, among other tasks, through interactions with viral and cellular proteins [49]. The structures of the *arenaviral* L–Z complexes aided in the understanding of the molecular mechanism of the on/off switching of vRNA production. The complexes of the Old World *arenavirus*, *Lassa virus* (PDB 7OCH, 7OE3, 7OE7, 7OEA, 7OEB, 7OJJ, 7OJK, 7OJL, 7OJN), and the New World *arenavirus, Machupo virus* (PDB 7CKM, 7ELC), revealed a single copy of the Z bound to the L polymerase at the interface of the PA subunit carboxy (C)-terminal-like (PA-C-like) regions and the RdRP (Figure 5) [50,51]. The central L-binding domain of the Z was shown to regulate the polymerase activity by associating with the motifs D and E, whereas the flexible extremities were proposed to interact with diverse proteins during the assembly for genome encapsidation. In the New World *Junin virus* (JUNV), the Z was found to interact with both the core-lobe and the palm domains of the L, inhibiting its activity during late infection (Figure 5) (PDB 7EJU) [52]. However, in the early stages, the lower concentration of Z in the cell and a conformational shift of the 680-loop due to interactions with host proteins such as eIF4E38 caused the Z to detach from the L, activating it [52].

### 2.2. Amino-Terminal Augmentations

The RdRPs of many viruses have additional domains at their N-terminus, such as MTases, endonucleases, cap binding domains (CBDs), and PRNTases, which play critical roles during RNA synthesis. 

The multifunctional polymerases of the family *Peribunyaviridae* are involved in both genome replication and transcription. Following replication, either vRNA or copies of the complementary RNA (cRNA) are generated. The 5′ caps from the host mRNAs are pirated with the help of the CBD and the endonuclease domains [22,53,54,55,56]. The previous crystal structure of the L protein of VSV suggested the threading of a ~20nt stretch of the template RNA through the polymerase domain during transcription [19]. Further, the disposition of the priming loop of VSV-L on the capping domain led to speculations on the coupling of capping and initiation during polymerization [19]. Nevertheless, the recent structures of the LACV polymerase in the pre-initiation and the elongation-mimicking states reveal the synchronized movement of the priming loop, the lid domain, and the mid-thumb ring linker to accommodate a ten-base-pair template–product duplex during elongation (Figure 6A) (PDB 7ORI, 7ORJ, 7ORK, 7ORL, 7ORM, 7ORN, 7ORO, 6Z6B, 6Z6G, 6Z8K) [55,56]. The authors suggested a properly orchestrated sequence of events, starting with the modification of a distal duplex promoter, followed by the generation of tension in the nucleotides 6–8 of the 3’ template, and finishing with the extension of the prime-and-realign loop for the polymerase to pursue active replication [55].

The RdRPs of the family *Orthomyxoviridae* are associated with the 3′ end of the genome where they are involved in initiating transcription soon after infection. The recent structures of the polymerases from this group have illustrated significant details on this phenomenon and the changes associated with cap-binding and polymerization. The structure of the FluA and *Influenza B virus* (FluB) polymerases revealed the ordering of the CBD (PB2, 320-489) and the “midlink” domain (251-319 of PB2 with the 490-536 linker) that mimicked the priming state of the polymerase when bound to their physiological substrate, the capped RNA (Figure 6B) (PDB 6EVK, 6EVJ, 6EUY, 6EUX, 6EUV, 6EVW) [57]. The CBD plays a pivotal role in directing the capped RNA to the endonuclease for cleavage and positioning the severed primer into the polymerase active site via a rotation of about 60° to commence transcription [20,58,59,60,61]. 

The *Phenuiviridae* RdRPs harbor an endonuclease domain at the N-terminal region followed by the motifs G-F-A-H-B-C-D-E that are located centrally. The large globular core includes the polymerase, the PA-C-like domain, the linker region, and the PB2-N-like domain. The recent cryo-EM reconstruction of the RdRP of *Severe fever with thrombocytopenia syndrome virus* (SFTSV) captured the structural details of the endonuclease domain of the enzyme in an early pre-initiation state of transcription (Figure 6C) (PDB 7ALP) [62]. The rotatable endonuclease domain of SFTSV L was housed back against the CBD (1695-1810), while in the LACV L it was flipped by ~180° along the long axis of the endonuclease facing the CBD [21,54]. This mandated substantial changes in the conformations of the SFTSV-L to initiate cap snatching. The crystal structure of the isolated putative CBD in complex with an m^7^GTP cap analog showed similarity to that of the *Rift Valley fever virus* (RVFV) CBD, comprising a long α-helix packed against a seven-stranded mixed β-sheet (PDB 6XYA, 6Y6K) [62]. The endonuclease domain and the CBD did not have a proper density in the cryo-EM structure of the RVFV L protein (PDB 7EEI). However, an integrated model constructed using SAXS and a low-resolution structure of the full-length L demonstrated that the endonuclease domain and the CBD were adjacent to one another, which is consistent with their functions. Similar to other bunyaviruses, the transcription of RVFV RNA required a primer through a cap-snatching process, although this was not the case with SFTSV [63]. The disposition of the priming loop of RVFV was also quite distinct from that of the SFTSV L and underwent large conformational shifts during transcription (Figure 6C). A unique feature of the RVFV RdRP was the ability to initiate de novo synthesis of RNA in the absence of 5′ vRNA, which was unseen in other *bunyaviral* RdRPs.

Amongst the members of the family *Flaviviridae*, the genera *Pestivirus* and *Hepacivirus* harbor a unique NTD of about 90 residues whose precise structure was unknown [17,64]. However, the structure of *Classical swine fever virus* (CSFV) RdRP revealed that the NTD forms an α/β fold with α-β-α-β-β-α arrangement showing extensive interactions with the polymerase (Figure 6D) (PDB 5YF5, 5YF6, 5YF7, 5YF8, 6AE4, 6AE5, 6AE6, and 6AE7) [65]. The authors speculate on the role of the flexible NTD in unwinding the RNA template and attribute fidelity to the RdRP [65].

The apo structures of the 3Dpols of the genera *Kobuvirus* (PDB 6R1I) and *Sicinivirus* (PDB 6QWT) reveal similarity to the *picornaviral* polymerases comprising the regular thumb, palm, and fingers subdomains and channels (Figure 6E) [66]. Using the crystal structures, subgenomic replicon assays, and phylogenetic analysis, it was established that the N-terminus of 3Dpol, liberated upon the proteolytic processing of the 3CD precursor, stabilizes the α10 (234-240) helix and consequently activates the picornaviral polymerases.

### 2.3. C-Terminal Bracelets 

The λ3 polymerases of *Reoviridae* are enormous cage-like structures that are encircled by their long N- and C-terminal extensions. While the N-terminal region wraps the continuous surface between the fingers and thumb subdomains of the polymerase, the C-terminal domain is an annular structure, popularly referred to as the “bracelet domain”, with a large opening capable of admitting ds RNA. 

The cryo-EM study using *Cytoplasmic polyhedrosis virus* (CPV) RdPR elucidated the crucial conformational changes associated with the thumb and the bracelet domains during three main stages of de novo transcription: (1) the transition from the quiescent to the initiation state brought on by the unwinding of the capped-terminal RNA; (2) the progress from the initiation to the early-elongation state via the insertion of the non-template RNA into its binding cleft; and (3) the switch from the early-elongation to the elongation state as a result of the template and non-template RNA annealing and the release of the nascent transcript (Figure 7A) (PDB 6TY8, 6TY9, 6TZ0, 6TZ1, 6TZ2) [67]. The conformational changes in the bracelet domain of the RdRP are dramatic and involve module A (912–1010) and module B (1067–1140). The former exhibits “relaxed” and “stiffened” states owing to changes in the turn number of an α-helix (955-1007), while the latter either “covers” or “uncovers” the transcript exit tunnel during the progress from quiescent to the elongation stages (Figure 7A) [67]. 

Pronounced conformational changes were also observable in the bracelet domain (779-1088) of rotaviral RdRP during transcription (Figure 7B) (PDB 6OGY, 6OGZ) [68]. The multifunctional bracelet domain aided the formation of the transcription bubble and directed the product RNA strands to exit during elongation. Additionally, the investigation emphasized the role of the N-terminal helix–loop–helix subdomain (31-69) in modulating RdRP activity through a switchable helix (349-360) of the capsid shell protein [68]. 

The in situ structure of the *Simian rotavirus* RdRP with triple-layer particles (TLPs) and double-layer particles (DLPs) in transcribing and non-transcribing modes revealed that localized conformational changes in the bracelet domain and the capsid shell protein facilitated the exit of the RNA transcript from the polymerase and the viral particle (PDB 6OJ3, 6OJ4, 6OJ5, 6OJ6) [69]. The cryo-EM and subparticle reconstruction of *Blue tongue virus* (BTV) RdRP in the TLPs and the DLPs provides insights into the role of the bracelet domain (936 to 1302) in the regulation and activation of the RdRP at the time of infection (Figure 7C) (PDB 6PNS, 6PO2) [70]. During entry, the movement of the VP3 protein, an inner layer protein of the viral subcore, disrupts the interactions of the protein’s N-termini with the RdRP at the five-fold axis. This forces a change of conformation of the bracelet domain, providing the requisite space and energy for the initiation of transcription [70].

### 2.4. Motifs, Pockets, and More

The non-canonical RdRPs of the family *Permutotetraviridae* display a unique cyclically permuted C–A–B–D connectivity of the four palm motifs [71,72]. The structures of the replicative complexes of *Thosea asigna virus* (TAV) RdRPs bound to the RNA template-primers were determined in three distinct replication states that provided high-resolution mechanistic details of the nucleotide addition cycle (Figure 8A) (PDB 7OM2, 7OM6, 7OM7, 7OM9, 7OMA) [73]. It was observed that the active site closed before NTP entry and the catalytic aspartate of motif A repositioned itself upon NTP and metal ion binding. The motif F underwent a major rearrangement as well, relocating by 20 Å in the apo state of the polymerase in relation to the RNA-bound form. The action mimics the conformational changes of motif F in flaviviral RdRPs and orients the ceiling of the NTP channel towards the catalytic cavity [73,74].

The structures of the intermediates during the forward and reverse translocation events of *Enterovirus* 71 (EV71) 3Dpol underpin the role of motif G in governing the rate-limiting step (Figure 8B) (PDB 6LSE, 6LSF, 6LSG, 6LSH) [75]. The motif G residues strongly resist the reverse translocation events by restricting the mobility of the RNA template and intermediates during translocation, thereby favoring the power stroke model of catalysis. Furthermore, a pocket in the fingers subdomain was found to be involved in a critical base-stacking interaction with the 5’ guanosine of the downstream RNA (PDB 6KWQ, 6KWR) [76]. The pocket comprises three distinct regions and includes the interacting residues His44, Arg277, Asn18, and Ser45, which are conserved in the RdRPs of *Picornaviridae*. 

Interestingly, the role of another pocket was highlighted in the high-resolution cryo-EM structures of the *Influenza A virus* (FluA) polymerase in association with the RNA (Figure 8C) [77]. The RNA adopted a loop-like conformation as it re-bound to a partially buried secondary site on the surface of the polymerase between PB1 and the PA-C following its exit. As a result, the 3′ end of the template remained tethered to the RdRP throughout the transcription process. This not only allowed the 3′ end to stay protected but also aided transcription restart by promoter reformation (PDB 6SZU, 6SZV, 6T0N, 6T0R, 6T0S, 6T0U, 6T0V, 6T0W, 6T2C, 6TU5, 6TW1) [77]. The same was reiterated with the polymerases of *Influenza D virus* (FluD) and A/H3N2 [28], where the 3′ end of the cRNA was bound to a secondary site on the polymerase [78,79]. The FluD polymerase structures in apo and promoter-bound form suggested the presence of two conformations (modes A and B) pertaining to the 3′ configuration of the promoter (Figure 8D) (PDB 6KUJ, 6KUK, 6KUP, 6KUR, 6KUT, 6KUV, 6KV5, 6KUU). While the mode A conformation was shown to promote the synthesis of cRNA, the mode B conformation mimicking the pre-initiation site was seen to be pivotal for vRNA synthesis [79]. Recently, the dynamics of the active site of the FluB polymerase during elongation were investigated using a modified template (Figure 8E) (PDB 6QCS, 6QCT, 6QCV, 6QCX, 6QCW) [60]. The study captured snapshots of the progressively extruding priming loop and widening active site cavity during the transition from the initiation to elongation steps. The co-crystal structure of the FluC polymerase with a Ser5-phosphorylated CTD peptide revealed the transcription-competent conformation of the polymerase in association with the CTD that facilitated the capture of the capped host mRNAs (PDB 6F5O, 6F5P, 6T0W) [80].

The crystal structure of the RdRPs of both the *Tick-borne encephalitis virus* (TBEV) (PDB 7D6N) [81] and the *Yellow fever virus* (YFV) (PDB 6QSN) (Figure 8F) [82] in association with their MTases showed similar structural features to the known RdRPs of other *flaviviruses*. Through mutational analysis, the authors identified a highly variable and solvent-accessible region in the nsp5 between the catalytic motifs B and C as the source of host-related diversity in the members of *Flaviviridae* [81]. 

## 3. Inhibitor Complexes of RdRP

RdRP has emerged as the most versatile protein for developing drugs against RNA viruses [4,14,83]. These drugs are broadly categorized into nucleoside and non-nucleoside inhibitors since they act by binding to the RdRP’s active site and allosteric regions, respectively. In the present study, the crystal structures of recently solved RdRP–inhibitor complexes (Table 2) have been analyzed for a greater understanding of their mode of binding and inhibition.

### 3.1. Picornaviral Inhibitors

Previously, *Picornaviridae* inhibitor complexes were shown to target antivirals at the interface of the fingers and the palm subdomains [4]. The EVD68 RdRP crystal structures in the apo state (PDB 6L4R) and in complex with the non-nucleoside-like inhibitor NADPH (PDB 5ZIT, Ligand ID NDP) revealed a binding pocket at the fingers subdomain close to the RNA template channel that correlated to a similar one in the Coxsackie virus B3 RdRP [84,85]. The binding of NADPH showed no apparent conformational variations with respect to the apo form and involved interactions with the residues Glu104-Asp107, Val206, Gly207, Ser284-Ser290, and Tyr322 (Figure 9A). The IC_50_ values of the NADPH against the RdRPs of EVD68, EVA71, and PV were determined to be 232.9 μM, 746.6 μM, and 342.8 μM, respectively. 

### 3.2. Coronaviral Inhibitors

Repurposing existing RdRP inhibitors for SARS-CoV-2 remains a viable option given the similarities in the essential drug-binding sites between the *Coronaviridae* members. A recent study of 22 SARS-CoV-2 RdRP inhibitors offered insights into the mechanics of viral RNA replication and a logical model for antiviral design [86]. The structures of the SARS-CoV-2 RdRP in complex with suramin and its derivatives (PDB 7D4F, Ligand ID H3U) revealed the presence of two distinct binding sites (Figure 10A) [36]. The first site (site 1) is located at the RdRP’s template binding region and involves the conserved motif G and the amino terminus of motif B. The naphthalene-trisulfonic acid moiety of suramin interacts with the residues Asn497, Lys500, Arg569-Lys577, Asn496, and Gly590 in site 1 and prevents the binding of the template (Figure 9B). Site 2 is positioned near the active site and is formed by the conserved motifs A, C, E, and F between the fingers and the thumb subdomains where the residues Arg550-Arg555, Arg836, and Asp865 participate in hydrogen bonding (Figure 9C). All the interacting residues are found to be conserved in the RdRPs of the family *Coronaviridae*. Suramin (IC_50_ 0.26 μM) and its derivatives are at least 20 times more potent than remdesivir against SARS-CoV-2 RdRP, but their use is limited because of poor cellular absorption caused by a strongly negative surface charge and off-target effects on helicases and polymerases [86]. 

The crystal structures of SARS-CoV-2 RTC with inhibitors favipiravir (PDB ID: 7DFG, Ligand ID 1RP) and ribavirin (PDB ID: 7DFH, Ligand ID RVN) are available from an unpublished work and reveal the binding of the inhibitors inside the RNA template channel. Favipiravir interacts with the residues Ser682, Asn691, and Lys545, while ribavirin interacts with the residues Asn691 and Ser682 via their carbamoyl and ribofuranosyl groups, and both show similar efficacy with IC_50_ of ~0.26 μM (Figure 9D,E, respectively). The cryo-EM reconstruction of the favipiravir ribonucleoside triphosphate (favipiravir-RTP) with the RdRP and the template-primer RNA duplex shows the binding of the inhibitor at the catalytic site of the RdRP in a non-productive conformation (PDB 7AAP, Ligand ID GE6) [87].

Recently, the mode of action of the highly potent drug remdesivir (Ligand ID GS-5734) was investigated by independent groups using cryo-EM (Figure 9F and Figure 10B) (PDB 7BV1, 7BV2) [86,88,89]. Yin et al. (2021) showed that the binding of remdesivir to the RTC along with the template-primer RNA (50 bp long) revealed significant conformational changes involving: (a) the movement of nsp7 with respect to nsp12; (b) the loop connecting the first two helices of the thumb subdomain; and (c) the residues of motif G. The inhibitor was covalently incorporated into the primer strand at the first replicated base pair, terminating the chain elongation [36]. Another study along the same line by Cramer et al. captured the structure of the remdesivir-stalled RdRP (PDB 7B3B, 7B3C, 7B3D). According to the authors, the cyano group of the inhibitor is sterically hindered in its transit past the ser861 side chain of the nsp12, resulting in a translocation barrier during elongation [88]. On the contrary, the structure of the elongation complex of SARS-CoV-2 with remdesivir by Gong and group revealed “i+3” delayed intervention of the inhibitor in meta-stable complexes, resulting in only a temporary pause of elongation (PDB 7DTE) [89]. The molecular mechanism underlying the inhibitory activity of molnupiravir (Figure 9G) (PDB 7OZU, Ligand ID 7OK, and PDB 7OZV Ligand ID 16B), the oral prodrug of beta-D-N4-hydroxycytidine (NHC) against SARS-CoV-2 RdRP, was recently elucidated [90]. When used in its active form as NHC triphosphates, molnupiravir is incorporated instead of CTP or UTP during the RNA synthesis. The structure of the RdRP–inhibitor complex demonstrated that the NHC can establish stable base pairs in the RdRP active center with either G or A, therefore explaining how the polymerase circumvents proofreading and creates altered RNA. 

### 3.3. Hepacivirial Inhibitors

The allosteric inhibitors of *hepaciviruses* and *flaviviruses* restrict interactions between the thumb and fingers subdomains, limiting catalysis and elongation [4]. In vivo studies of HCV RdRP in association with GSK5852 (PDB 4KHM, Ligand ID 1PV) is the first experimentally validated palm II NS5B non-nucleoside inhibitor that has broad-spectrum antiviral activity owing to its N-benzyl boronic acid moiety (IC_50_ ~ 50nM). Although strong in action, this inhibitor requires a large daily dose for effectiveness due to its short plasma half-life (5 h) in human subjects [91]. An earlier work by Chong et al. (2019) sought to find second-generation palm II NS5B inhibitors that prevent the inhibitor’s benzylic oxidation, preserving its efficacy while increasing its plasma life [92]. These include compounds 25 (PDB 6MVK, Ligand ID K4J), 31 (PDB 6MVQ, Ligand ID K4M), 45 (PDB 6MVP, Ligand ID K4S), and 49 (GSK8175, PDB 6MVO, Ligand ID K4P). Many of these inhibitors form extensive water-mediated hydrogen bonds with the polymerase except compound 25, which forms direct hydrogen bonds with Arg200 and Tyr448 via its oxazolidinone group and with Asn316 and Ser365 through the methyl carbamoyl moiety (Figure 11A). The water-mediated hydrogen bonds are formed by the N-phenyl group of compound 45 with residues Arg200, Tyr448, Asn316, Arg386, and Tyr415 (Figure 11B) and the triazole moiety of compound 31 with residues Arg200 and Tyr448 (Figure 11C). The oxaborole group of compound 49 replaces the boronic acid of compound 31 and contributes to two of the six water-mediated hydrogen bonds with residues Gly449, Arg386, Asn411, Ser365, and the conserved Arg200 (Figure 11D). Compound 49, a sulfonamide-N-benzoxaborole analog, was the most potent inhibitor amongst all the screened derivatives; it showed broad-spectrum activity with a long plasma half-life of about 60-63 h and IC_50_ of 16 μM in humans. 

The co-crystal structures of HCV RdRP with N-ethylmethylsulfonamide analogs, compound 5 (PDB 5QJ0, Ligand ID J6D) and compound 13 (PDB 5QJ1, Ligand ID J6J), are available from a recent drug screening study that was aimed at solving the P-glycoprotein and BCRP transporter-mediated efflux problems of benzofuran-derived palm site allosteric inhibitors [93]. Both derivatives exhibit comparable binding patterns analogous to compound 49, but the former showed enhanced pharmacokinetic properties and oral bioavailability in studies with rat models. In the crystal structures of the complexes, compound 13 orients towards Gly449, Arg386, and Asn411 due to the presence of an oxaborole moiety, while compound 49 tilts in the opposite direction towards Arg200 and Tyr195 due to interactions with the carbamoyl group in the binding pocket (Figure 11E,F). Compound 49 was potent against genotype (GT) 1, while compound 13 showed efficacy against GT 2a owing to a hydrogen bond between the ethylamino moiety and the Gln414 of NSP5 [93]. 

### 3.4. Flaviviral Inhibitors

Recently, the crystal structures of DENV3 RdRP with two non-nucleoside inhibitors, NITD-434 (PDB 6XD0, Ligand ID 6V0M) and NITD-640 (PDB 6XD1, Ligand ID 6V0J), with moderate activity against the polymerase were reported [94]. The IC_50_ values ranged from 6 to 17 μM in DENV4 and 0.9 to 13 μM in DENV1 RdRPs. The NITD-434 binds inside the RNA channel, displacing Val603 in motif B, whereas the NITD-640 orders the loops of the fingers domain preceding motif F near the tunnel entrance upon binding (Figure 12A,B). Most of the interacting residues in the binding pockets of the two compounds are conserved in other *flaviviruses* such as *Japanese encephalitis virus*, YFV, *West Nile virus*, and *Zika virus* [94].

The structures of the DENV2 and DENV3 polymerases in their apo states and with their inhibitor (Compound RK-0404678, Ligand ID B5C) were solved using X-ray crystallography (PDB 6IZX, 6IZZ) [95]. Similar to the SARS-CoV2-RdRP, there are two binding sites for the DENV2 polymerase with an IC_50_ of ~6.0 μM. The benzothiazole ring-mediated interactions of the inhibitors were observed with the residues Arg773, Asn777, Cys780, Met809, Trp833, Tyr882, and Met883 in both DENV2 and DENV3 polymerases at site 1, whereas the residues Val507-Gly511, Ser661, and Cys709 were involved at site 2 (Figure 12C,D) [95]. 

The derivatives of the NIT29 inhibitor (Ligand ID KY3) that showed preferential binding to an allosteric pocket between the thumb and palm subdomains (PDB 6LD1, 6LD2, 6LD3, 6LD4, 6LD5) were used to screen for potential non-nucleoside inhibitors of Zika viral RdRP [96]. Fluorescence-based alkaline phosphatase-coupled polymerase assays (FAPAs) for initiation and elongation revealed inhibitory concentrations of 44.8 μM and 51.9 μM, respectively. In a cell-based experiment, compounds 13, 14, and 15 possessing the scaffold and aryl sulfonamide moiety displayed modest activity against RdRP and interacted with the residues Arg739, Trp797, Arg731, and Thr796 (Figure 12E–H). Compound 7, comprising the thiophene group and the acid moiety from the phenyl ring, showed no activity [96]. 

## 4. Exploring NADPH as a Broad-Spectrum Inhibitor

The in vitro titration of the polymerases of EVD68, EV71, and Poliovirus with NADPH resulted in decreased activity where the ligand acted as an allosteric inhibitor rather than a chain terminator during replication [84]. Furthermore, the crystal structure of the EVD68–NADPH complex showed that the interacting residues were conserved in all enteroviruses. This led us to speculate that the ligand could have similar binding and inhibitory properties in the RdRPs of other viruses as well. Hence, we performed blind docking of NADPH (Ligand ID NDP) against RdRPs of select viruses in the apo and template-bound forms using AutoDock Vina 1.1.2. These include HCV (PDB 1NB4, 1NB7), SARS-CoV-2 (PDB 7BTF, 7C2K), *Poliovirus* (PDB 4K4W, 4R0E), and EV71 (PDB 6KWQ, 3N6L). After protein preparation and removal of water molecules, the ligand-based docking was carried out with default parameters using a genetic algorithm. The NADPH showed a striking similarity in the binding poses that depended solely on the occupancy of the template channel of the RdRP. In the apo structures, the NADPH was bound inside the template channel closer to the pinky of the fingers subdomain, whereas in the template-bound elongation complexes, the ligand was displaced by more than 5 Å away from its original position and towards the thumb subdomain owing to the binding of the RNA (Figure 13). The interaction energies of the NADPH with the RdRP in the apo states were consistently lower than the corresponding elongation complexes in all viruses owing to additional interactions of the ligand with the RNA (Table 3). Intriguingly, the ability of NADPH to effectively engage with the polymerase at different phases of replication is demonstrated by the changes in binding sites. As suggested in the earlier structural and in vivo studies with EVD68 [84], the NADPH binding in other viral RdRPs could exert comparable effects, impairing translocation and catalysis, thus proving its suitability to be developed as a broad-spectrum allosteric inhibitor. 

## 5. Conclusions and Future Directions

Recent investigations, largely cryo-EM-based, have revealed structural details of the RdRPs bound with cofactors, nucleic acid moieties, analogs, and inhibitors, enhancing our understanding of their mechanism of action. Although the fundamental structure of RdRP has stayed constant over time, there are variations at the individual level that remain unexplored. The association of RdRPs with cofactors that are either self-encoded or derived from the host facilitates the timing of genome replication and transcription. For instance, the in situ structures of *reoviral* RdRPs demonstrated that the disruption of capsid protein interactions with RdRP during disassembly provided the essential room and energy for transcription to start [69,70]. Likewise, the proteolytic processing of the 3CD of *picornaviruses* [66] and the association of VPg in *caliciviruses* [27] aided in the activation of the respective RdRPs. The binding of the P with the L proteins in *mononegaviruses* [46,47], and the nsp7-9, nsp13, nsp14, and nsp16 with nsp12 in SARS-CoV-2 [38,97], promoted active replication, while the Z protein of the *arenaviruses* [52] inhibited the synthesis of vRNA. The RdRP complexes with RNA or its analogs offer previously unknown details of structure–function relationships. For example, the re-association of the 3′ template RNA at the polymerase surface of FluA following exit protected it against nucleases [77], and the interactions of the NTDs of the *pestiviral* RdRPs promoted the unwinding of the RNA duplex [65]. The varied nature of the inhibitor binding pockets of RdRPs from distinct viruses necessitates the search for antivirals with unique pharmacophoric properties [14,98]. For example, the benzofuran-, thiazolidinone-, or anilinobenzothiazole-derived inhibitors were effective against HCV, while pyrazole or bibenzylsubstituted piperazine-containing compounds performed well against Zika and the octahydroquinazoline derivatives strongly inhibited DENV RdRP. In addition, our in silico investigations using NADPH demonstrated the possibility of using it as a broad-spectrum antiviral after experimental verifications. The comparison of the catalytic complexes of representative viral RdRPs, including SARS-CoV-2, revealed the broad-spectrum effectiveness of nucleotide analogs with 1’ modifications as potential antivirals [89]. Curiously, the SARS-CoV-2 oral drug molnupiravir induced widespread mutagenesis in the viral genome, presenting an unusual antiviral strategy [90]. Thus, the effective modifications of the current antivirals and the discovery of new ones should eventually result from a comprehensive understanding of the dynamics of polymerization, mandating additional research. The end is just the beginning.

## Figures and Tables

**Figure 1 viruses-14-02200-f001:**
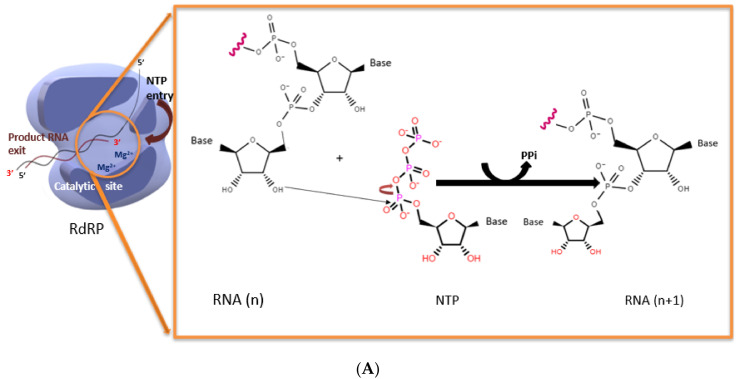
(**A**) Polymerization reaction catalyzed by RdRP. (**B**) The structure of poliovirus RdRP (PDB 1RA6) showing the fingers (magenta ribbon), palm (gray), and thumb (green) subdomains. The motifs A (red), B (orange), C (yellow), D (navy blue), E (cyan), F (blue), and G (brown) are indicated. The aspartates involved in catalysis from motifs A and C are shown. The conserved aspartates of the motif A (D-x(4,5)-D) and motif C (GDD) are represented as ball-and-stick, and the main channels are indicated.

**Figure 2 viruses-14-02200-f002:**
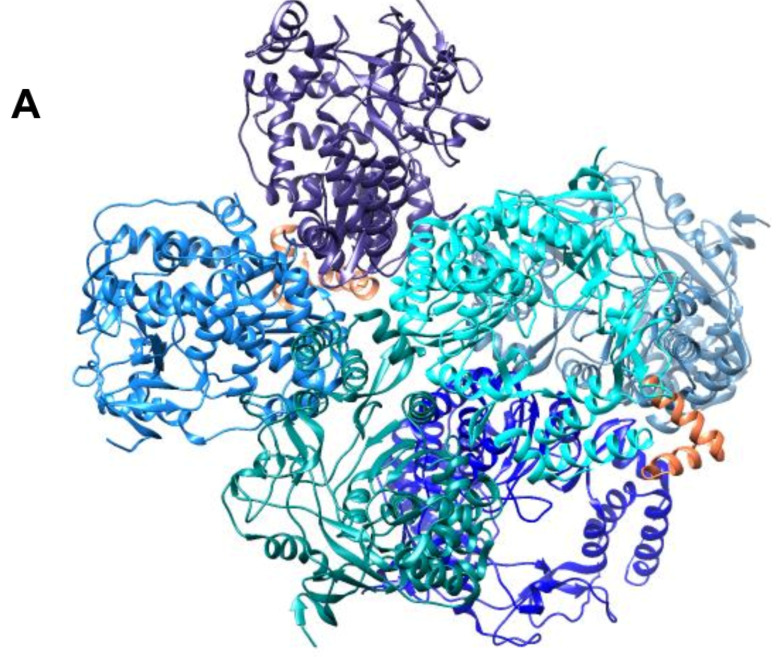
The RdRP–VPg complexes of *caliciviruses*. (**A**) Multimerization of MNV RdRP–VPg complex. Six monomeric units of RdRP of MNV (PDB 5Y3D), shown in shades of blue, associate with each other and the VPg moiety (shown as orange ribbons) to form efficient ball-like replication machinery. (**B**) Zoomed-in view of a single MNV RdRP–VPg interaction unit. The close-up of the two molecules of MNV RdRP is shown in shades of blue with the interacting helices of VPg (α1 and α2) shown in orange. The residues of the RdRP participating in the interaction with the VPg are also shown in ball-and-stick representation and colored green.

**Figure 3 viruses-14-02200-f003:**
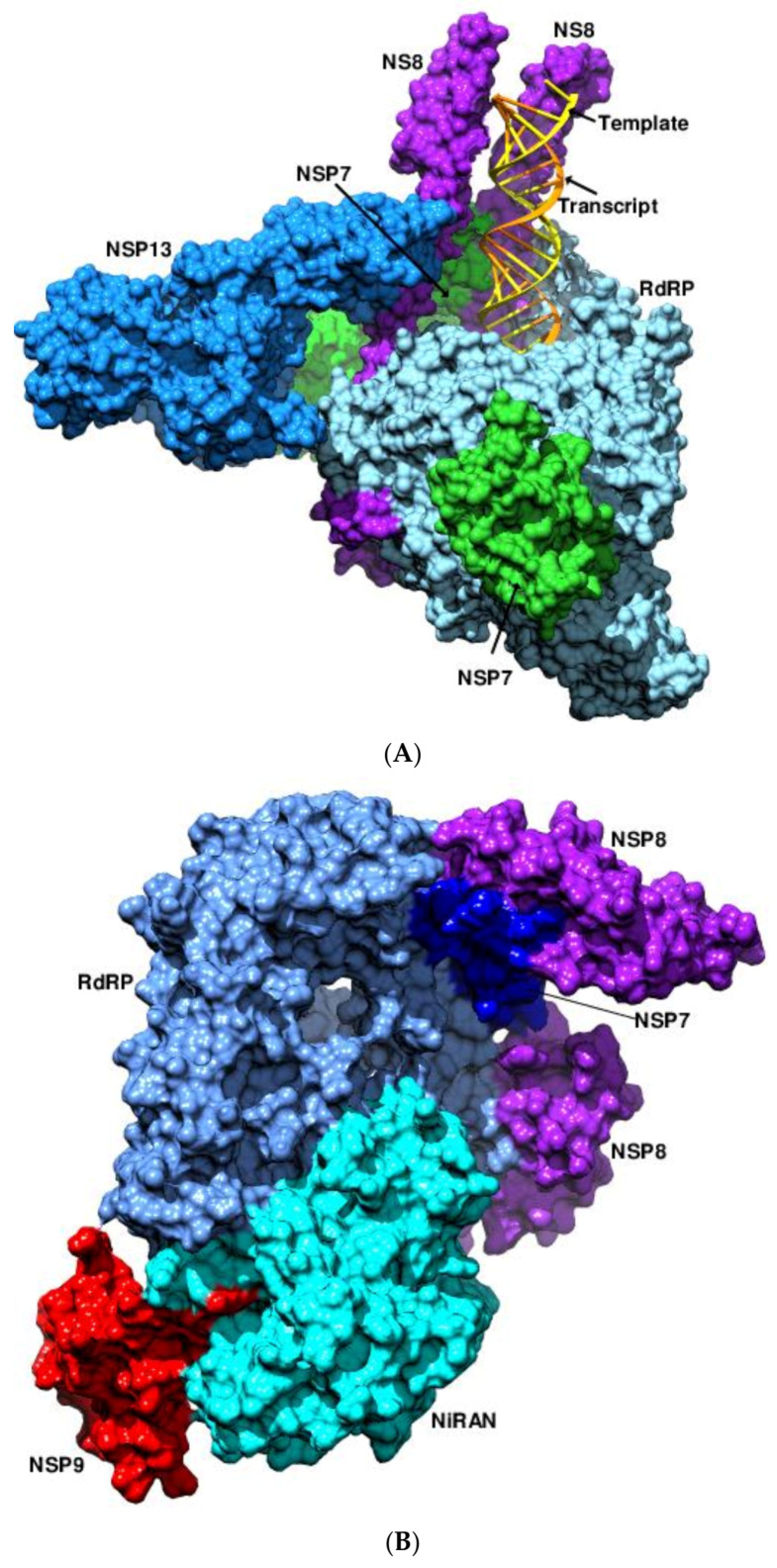
RTCs of SARS-CoV-2. (**A**) RdRP (light blue), nsp7 (green), nsp8 (purple), nsp13 (Dodger blue), template (yellow), and product (orange) RNA are shown in surface representation. The positively charged tentacle-like amino-terminal extensions of the two nsp8 copies can be seen clasping the RdRP. The disposition of the nsp12 with respect to the other domains indicates its extensive interaction with the RTC. (**B**) The complex of the nsp9 with the RTC is shown. The interaction of the NiRAN domain (1-385, cyan) with the nsp9 (red surface) is implicated as one of the first steps in the capping of the 5′ genomic RNA. (**C**) The NTD of STAT2 (1-136, dark green) interacts with the residues 314-327 of the fingers subdomain (305-495, navy blue), while the coiled-coil domain of STAT2 (137-315, lime green) interacts with residues 847-855 of the thumb subdomain (721-888, Dodger blue). The NTD (273-307, purple), and the palm (496-720, cyan) subdomains of RdRP are indicated.

**Figure 4 viruses-14-02200-f004:**
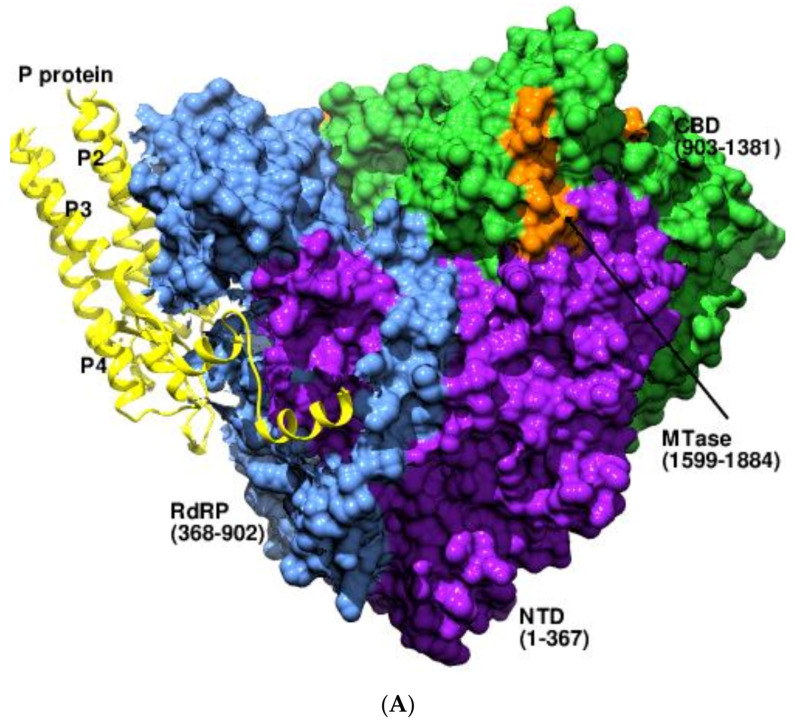
The association of the P domain of HPMV with the L. (**A**) The figure shows the interaction of the coiled-coil domain of the P (residues 169-195) with the L protein (PDB 6U5O). The four helices of the tetrameric P domain are shown as yellow ribbons, and the important domains of L are indicated in different colors in surface representation. (**B**) The superposed structure of HRSV (PDB 6PZK, blue ribbon), PIV5 (PDB 6V85, gray ribbon), and HPMV (PDB 7ORL, green ribbon) RdRPs and their corresponding P domains. The strong resemblance between the structural conformations of HPMV and HRSV and their association with P is observable. The large conformation shift of the MTase, CBD, and CTD of PIV5 with respect to the mononegaviral L–P complex is obvious, though the P interactions seem similar. (**C**) Cooperative binding of ANPP32 to two copies of FluC polymerase during replication (PDB 6XZQ). The structures of human FluC polymerases (RdRP: brown, PA: green, PB: orchid, RNA: yellow) bound to the amino-terminal leucine-rich repeat domain of ANP32A (blue) are shown. The ANPP32 plays a pivotal role during replication by stabilizing the replication complex.

**Figure 5 viruses-14-02200-f005:**
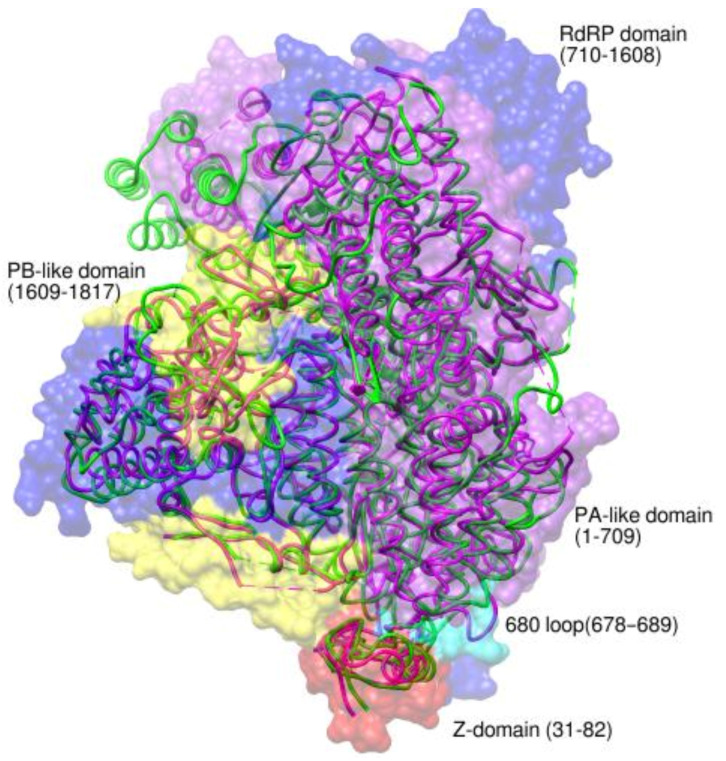
Superposition of Junin, LACV, and MACV L–Z complexes. The Junin RdRP complex showing the PA-like (purple), RdRP (blue), and PB2-like (yellow) domains. The PA-like domain harbors the endonuclease (residues 1-196). The Z-protein that acts like a molecular switch is shown in red color while the loop-680 that binds to the host factors is shown in cyan. The superposed structures of the L and Z proteins of LASV (green ribbons) and MACV (pink ribbons) indicate variability at the N-terminus.

**Figure 6 viruses-14-02200-f006:**
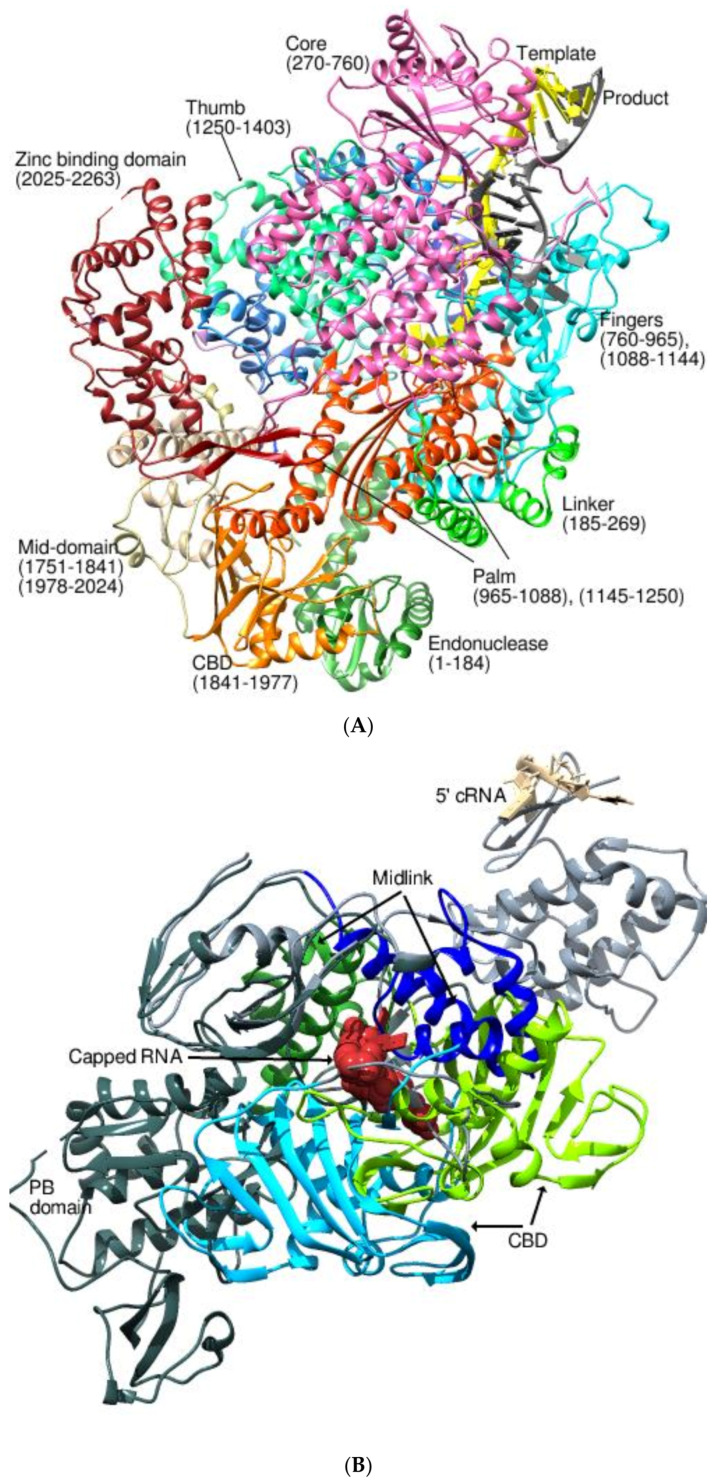
The contributions of the NTDs in replication and transcription. (**A**) NTDs of the *peribunyaviral* polymerases. The transcription initiation stage of the LACV polymerase (PDB 7ORL). The endonuclease and the CBD are shown in sea green and light orange colors, respectively. The other domains from N- to C-terminus are depicted in rainbow colors and represented as ribbons. The template and the product RNA are shown in yellow and gray colors, respectively. The domains at the amino terminus are involved in cap-snatching. (**B**) NTDs of the Flu polymerases. The structure of FluA RdRP (PDB 6EAJ, light gray) with the bound physiological RNA primer (maroon spheres) superposed on the apo state of FluB RdRP (PDB 5EPI, dark gray) with bound 5′cRNA (Tan). The CBDs (320-489) of FluA and FluB are colored in light blue and light green colors, respectively, while the midlink (251-319) regions of the two are indicated in dark blue and dark green colors, respectively. The CBD and the midlink regions have rotated by 60° and 77°, respectively, with respect to the corresponding domains in the apo state of FluB. (**C**) NTDs of *Phenuiviridae* RdRPs. The superposed ribbon representation of the L protein of SFTSV (PDB 6XYA) on the L of RVFV (PDB 7EEI). The domains of SFTSV are shown in rainbow colors while the RVFV is shown in gray. The marked displacement of the helix (390-400) in RVFV is obvious. The endonuclease, CBD, b, and lariat domains are missing in the RFVF structure. (**D**) NTDs of the *flaviviral* RdRPs. The nsp5 of CSFV showing the flexible NTD (1-86), the linker (87-92), and the RdRP domains. The structures of PDB 5Y6R (light blue) and 5YF6 (dark blue) show the NTD in a similar disposition, while those of 6AE7 (green) and 5YF8 (yellow) are flipped by almost 180° and appear in an extended conformation. (**E**) NTDs of the *picornaviral* RdRPs. The structures of kobuviral and siciniviral RdRPs. The *kobuviral* (blue) (PDB 6R1L) and *siciniviral* (orange) (PDB 6QWT) 3D^pol^ superposed and depicted in ribbon representation showing the position of α10 that is implicated in the activation of the polymerase.

**Figure 7 viruses-14-02200-f007:**
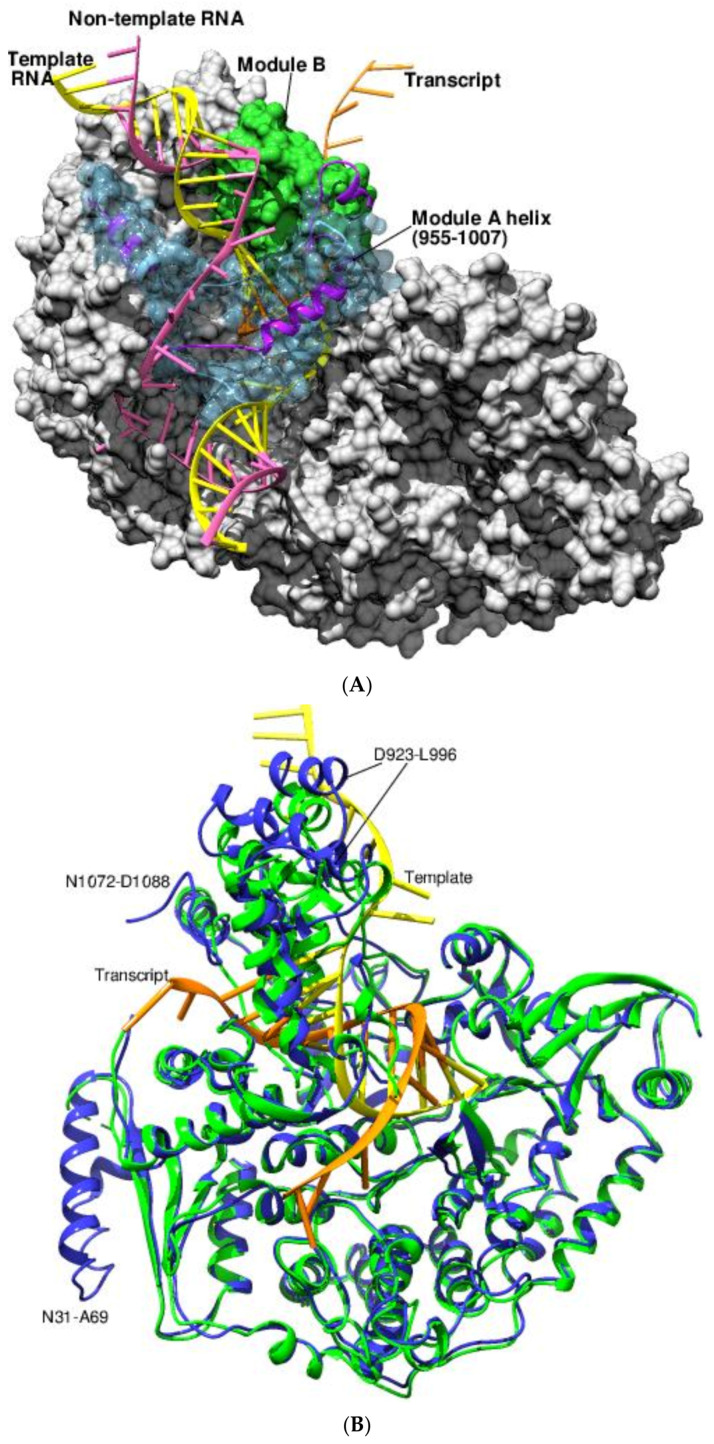
The RdRPs of *Reoviridae.* (**A**) The figure shows the RdRP domain of CPV in the quiescent stage with the bound RNA during transcription. The template (yellow), non-template (pink), and transcript (orange) RNA are shown as ladders. Modules A (912-1010, light blue) and B (1067-1140, green) from the polymerase (PDB 6TY8, gray surface) are indicated. The noticeable deflection of the module A α-helix (955-1007) of the C-terminal bracelet domain during the elongation stage (PDB 6TZ2, purple ribbon) is evident. (**B**) The in situ structure of rotaviral RdRP. The RdRP of *Rotavirus* at transcript-elongated state (blue ribbon, PDB 6OGZ) was superposed on the duplex open state (green ribbon, PDB 6OGY). The marked conformational changes around residues N31-A69, D923-L996, and N1072-D1088 can be observed. The transcript is shown in orange and the template strand in yellow. (**C**) The in situ structure of BTV RdRP. The BTV RdRP in its core (PDB 6PO2) shows the disposition of the VP3 (green ribbon) and the RdRP (blue surface). The close interaction of the N-terminal region (red ribbon) of the alternating VP3 with the bracelet domain of the RdRP (purple) around the five-fold axis is believed to be responsible for the activation of the RdRP during infection. E19, D21, and D26 (yellow sticks), three negatively charged residues on VP3’s N-terminal loop, interact with the positively charged region of the bracelet domain.

**Figure 8 viruses-14-02200-f008:**
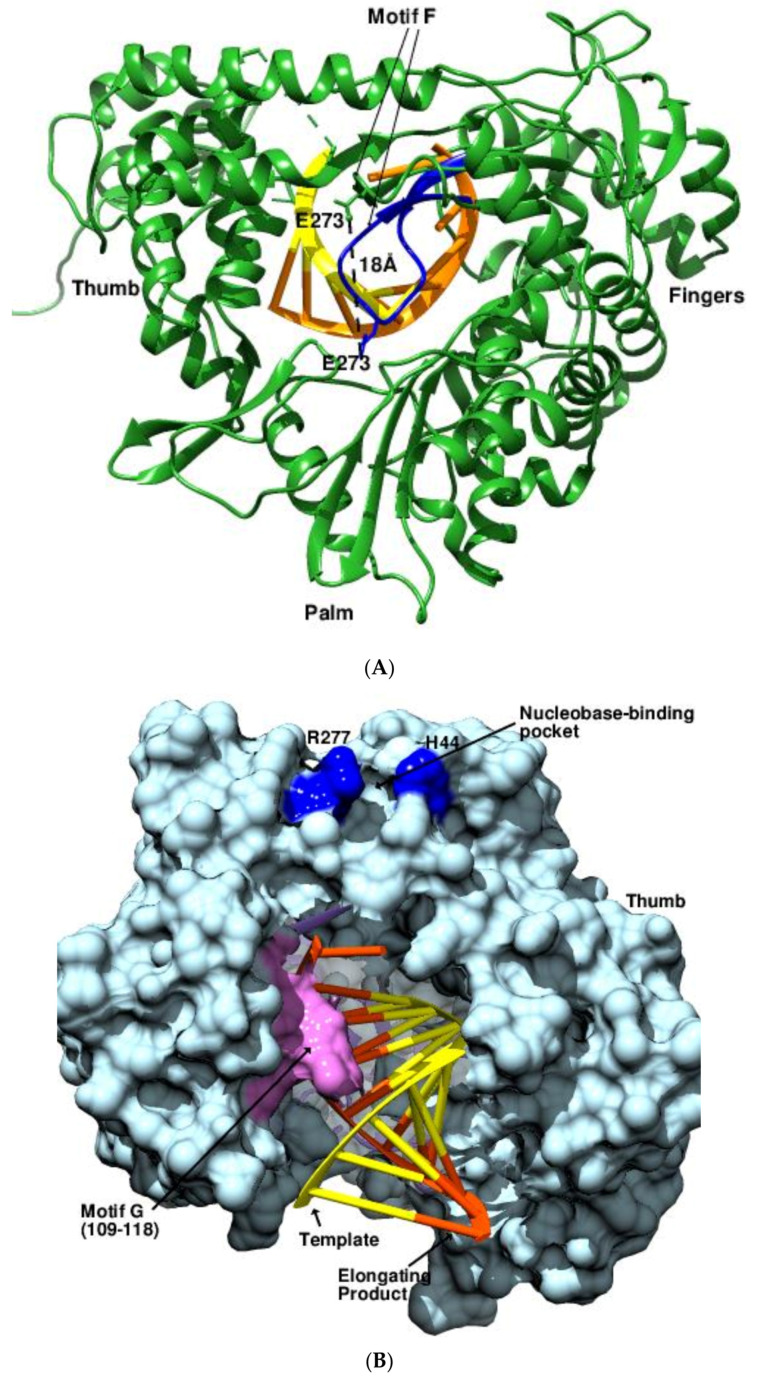
Motifs, pockets, and their role in replication. (**A**) The configuration of the motif F in the apo and the template-bound states of TAV RdRP. The reorganization of motif F in the bound (green, PDB 7OM6) and unbound (blue, PDB 7OM9) states is highlighted. The template and the product strands are shown in yellow and orange, respectively. (**B**) *Picornaviral* EV71 RdRPs at the translocation start stage (PDB 6LSE, light blue surface) with the template (yellow) and product (orange) RNAs represented as a ladder. The key interacting residues R277 and H44 of the nucleobase-binding surface pocket are shown in blue. The motif G (109-118), implicated in resisting the reverse translocation events, is colored in pink. (**C**) The structures of FluA RdRP showing the disposition of the template, product, and the capped mRNA during elongation (PDB 6T0V) and termination (PDB 6TW1). The PB (gray surface) and RdRP (light blue ribbon) domains are indicated, while the PA domain is masked for clarity. The template RNA is depicted in light yellow (elongation) and yellow (termination), the product in orange (elongation) and red (termination), and the capped mRNA in magenta (elongation) and purple (termination) colors. (**D**). The varied dispositions of the FluD in mode A (PDB 6KUK) and mode B (PDB 6KUR) conformations bound to the viral RNA. The former is known to mimic the pre-initiation conformation, facilitating RNA binding, while the latter adopts a conformation promoting the synthesis of cRNA. (**E**) Active site rearrangement of FluB RdRP during the replication. The superposed RdRP domains of the initiation (PDB 6QCS) in cyan color and elongation (PDB 6QCT) complexes of FluB in blue showing the extruding priming loop (187-204). The huge displacement of ~20 Å of the priming loop is obvious. The template RNAs are shown in yellow, the product RNAs are shown in orange, and the capped mRNA is shown in magenta and red. (**F**) The structural superposition of the RdRPs of TBEV (PDB 7d6n, orange ribbon) and YFV (PDB 6qsn, blue ribbon and surface) showing the varied disposition of the MTases. In TBEV, the MTase structure was unresolved. The pocket in the BC region (616-658, red ribbon) of the palm subdomain, which is implicated in host-related diversity of flaviviruses, is highlighted.

**Figure 9 viruses-14-02200-f009:**
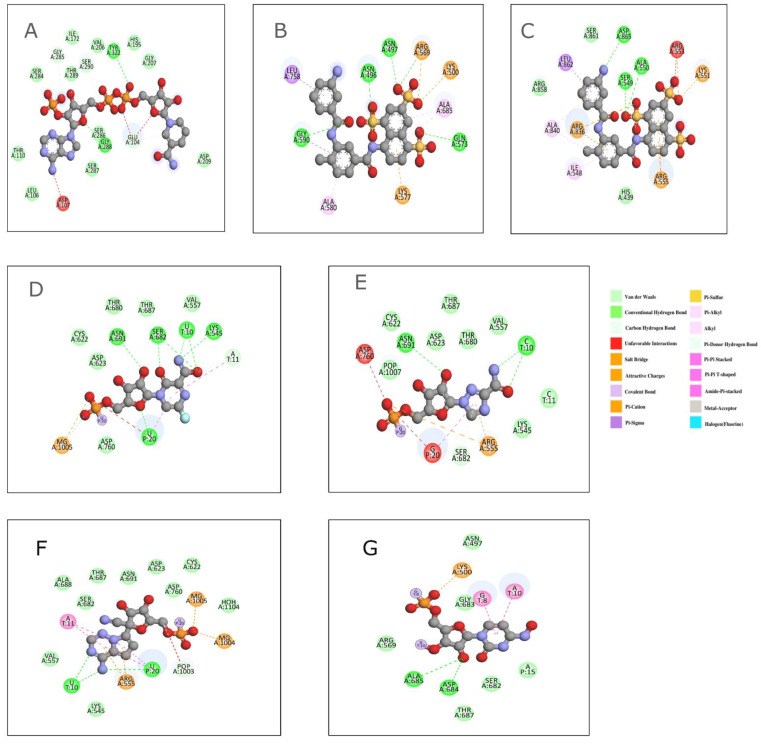
The ligand–RdRP interaction diagram for the Enterovirus and SARS-CoV-2 RdRPs with their inhibitor compounds. (**A**) Enterovirus D68 RdRP–NADPH (PDB 5ZIT), (**B**) SARS-CoV-2 RdRP–suramin (site 1) (PDB 7D4F), (**C**) SARS-CoV-2 RdRP–suramin (site 2) (PDB 7D4F), (**D**) SARS-CoV-2 RdRP–favipiravir (PDB 7DFG), (**E**) SARS-CoV-2 RdRP–ribavarin (PDB 7DFH), (**F**) SARS-CoV-2 RdRP–remdesivir (PDB 7BV2), (**G**) SARS-CoV-2 RdRP–molnupiravir (PDB 7OZU).

**Figure 10 viruses-14-02200-f010:**
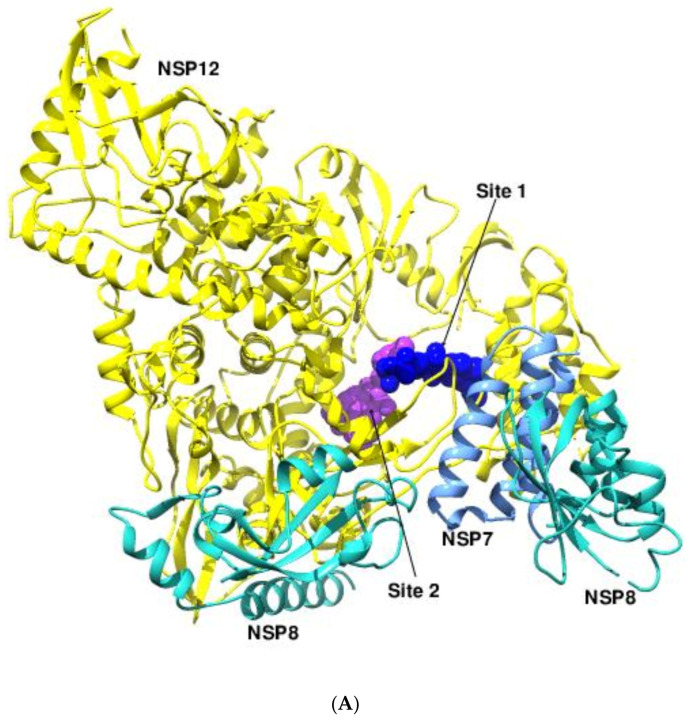
Inhibitor binding sites of DENV2 and SARS-CoV-2 RdRPs. (**A**) The binding sites of suramin derivatives in SARS-CoV-2 RdRP (yellow ribbon). The first one is near the active site (magenta), and the other site is located at the template binding region (blue). The NSP7 and NSP8 that are bound to the RdRP are shown in shades of blue. (**B**) The structure of SARS-CoV-2 RdRP with the bound nucleoside-analog remdesivir (green ball and stick). The template and the product are shown in yellow and orange, respectively. Ser861 obstructs the cyano group of the inhibitor during translocation. (**C**) The structure of the DENV2 RdRP showing the interactions with the inhibitor RK-0404678 at the active site (magenta) and in the thumb subdomain (blue). The inhibitor is shown as spheres and the RdRP is depicted as a yellow ribbon.

**Figure 11 viruses-14-02200-f011:**
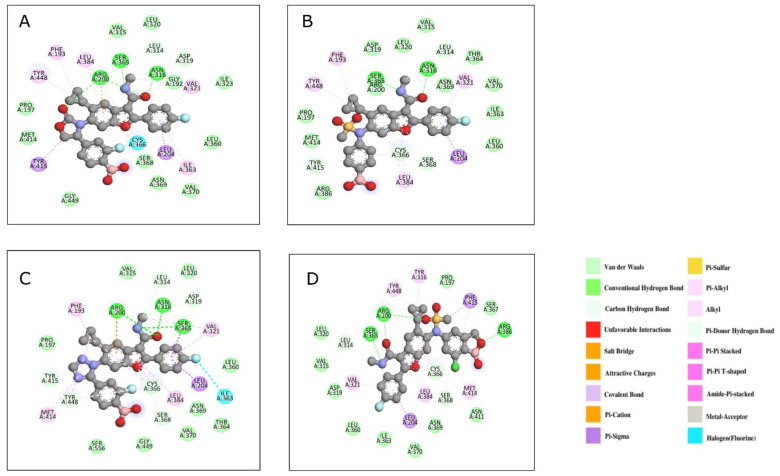
The HCV RdRP–inhibitor complexes showing the interactions of different residues with the bound inhibitor: (**A**) compound 25 (PDB 6MVK), (**B**) compound 45 (PDB 6MVP), (**C**) compound 31 (PDB 6MVQ), (**D**) compound 49 (PDB 6MVO), (**E**) compound 5 (PDB 5QJ0), and (**F**) compound 13 (PDB 5QJ1).

**Figure 12 viruses-14-02200-f012:**
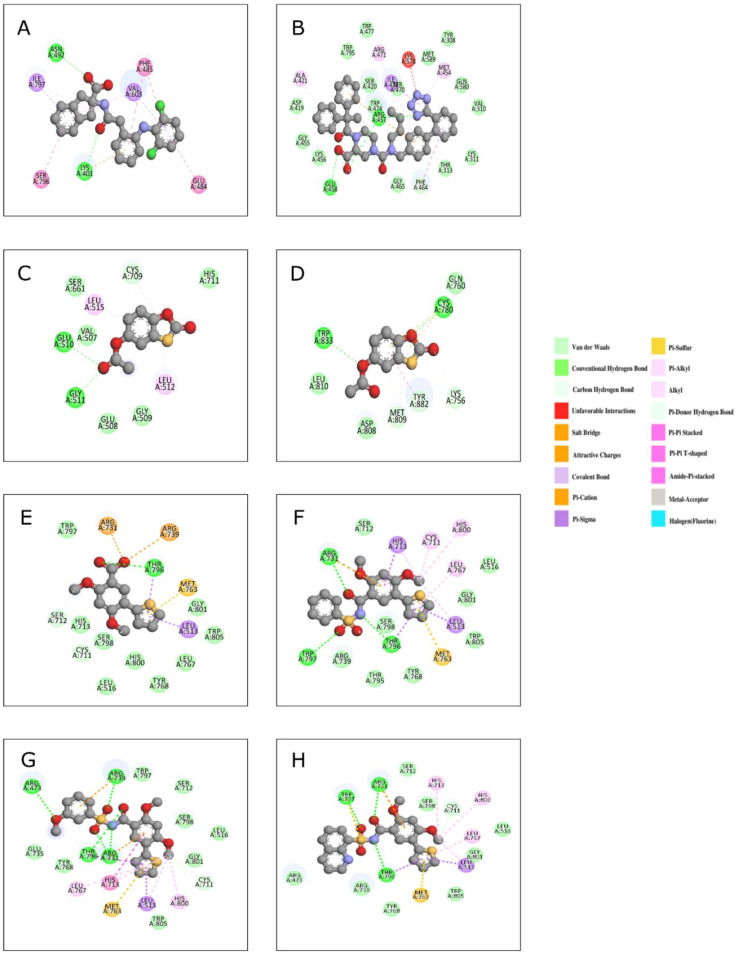
The DENV and Zika RdRP inhibitor complexes showing the interactions of different residues with the bound inhibitor: (**A**) DENV3 RdRP–NITD-434 (PDB 6XD0), (**B**) DENV3 RdRP–NITD-640 (PDB 6XD1), (**C**) DENV2 RdRP–RK0404678 (PDB 6IZX), (**D**) DENV2 RdRP–RK0404678 (PDB 6IZZ), (**E**) Zika virus RdRP–compound 7 (PDB 6LD2), (**F**) Zika virus RdRP–compound 13 (PDB 6LD3), (**G**) Zika virus RdRP–compound 14 (PDB 6LD4), (**H**) Zika virus RdRP–compound 15 (PDB 6LD5).

**Figure 13 viruses-14-02200-f013:**
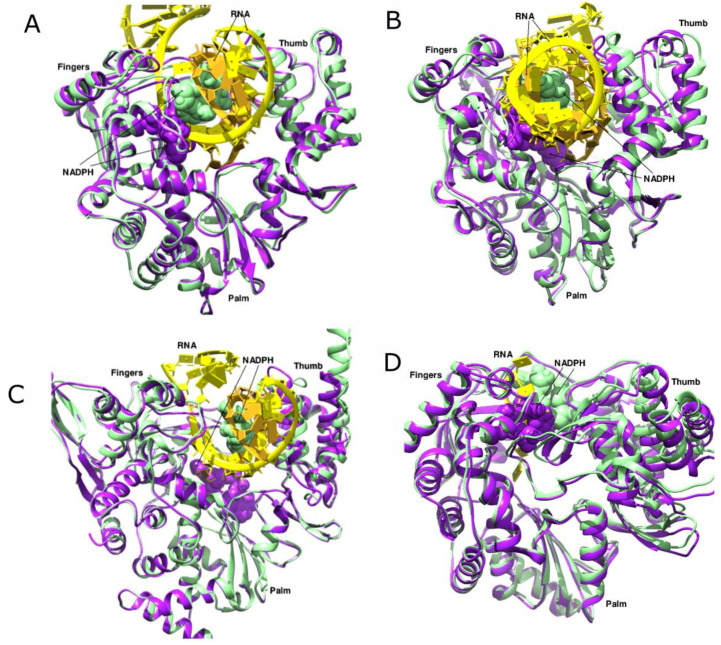
In silico docking of RdRP in the apo and template-bound states with NADPH. The apo form is shown in purple and the RNA-bound form is depicted in green. The polymerase is shown in ribbon representation and the ligand is shown as spheres. The RNA template and product are shown in yellow and golden colors, respectively. (**A**) EV71 (PDB 6KWQ, 3N6L) with NADPH, (**B**) PV (PDB 4K4W, 4R0E) with NADPH, (**C**) SARS-CoV-2 (PDB 7BTF, 7C2K) with NADPH, (**D**) HCV (PDB 1NB4, 1NB7) with NADPH.

**Table 1 viruses-14-02200-t001:** Recent structures of viral RNA-dependent RNA polymerases (RdRPs).

Family	Genus	PDB ID	Citation
*Phenuiviridae*	Rift valley fever virus (RVF)	7EEI	10.1128/JVI.01713-21
	Dabie bandavirus/SFTS virus (SFTSV)	7ALP 6Y6K,6XYA	10.1038/s41564-021-00901-3 10.1093/nar/gkaa253
*Arenaviridae*	Lassa mammarenavirus (LASV)	7OCH, 7OE3, 7OE7, 7OEA, 7OEB, 7OJJ, 7OJK, 7OJL, 7OJN	10.1038/s41467-021-27305-5
	Machupo mammarenavirus (MACV)	7ELC 7CKM	10.1038/s41564-021-00916-w 10.1038/s41564-021-00916-w
	Junin virus (JUNV)	7EJU	10.1038/s41467-021-24458-1
*Pneumoviridae*	Human metapneumovirus (HMPV)	6U5O	10.1038/s41586-019-1759-1
	Human respiratory syncytial virus A2 (hRSV)	6PZK	10.1016/j.cell.2019.08.014
*Peribunyaviridae*	La Crosse virus (LACV)	6Z6B, 6Z6G, 6Z8K	10.1038/s41467-022-28428-z 10.1038/s41467-020-17349-4
*Paramyxoviridae*	Parainfluenza virus 5(PAV5)	6VAG	10.1073/pnas.1919837117
*Orthomyxoviridae*	Influenza A (IAV)	6EVK, 6EVJ, 6EUY, 6EUX, 6EUV, 6EVW 6SZU, 6SZV, 6T0N, 6TU5, 6T0S, 6T0U, 6T2C, 6T0R, 6T0V, 6T0W 6TW1, 6QNW, 6QPF, 6QPG, 6QWL, 6QX3, 6QX8, 6QXE, 6RR7	10.1093/nar/gkx1210 10.1016/j.cell.2020.03.061 10.1038/s41586-019-1530-7
	Influenza B (IBV)	6QCS, 6QCT, 6QCV,QCX, 6QCW 6QNW, 6QPF, 6QPG, 6QWL, 6QX3, 6QX8, 6QXE, 6RR7 6F5O, 6F5P	10.1038/s41594-019-0232-z 10.1038/s41586-019-1530-7 10.1016/j.molcel.2018.05.011
	Influenza C (ICV)	6XZD, 6XZQ, 6Y0C, 6XZG, 6XZP, 6XZR	10.1038/s41586-020-2927-z
	Influenza D (IDV)	6KUJ, 6KUK, 6KUP, 6KUR, 6KUT, 6KUV, 6KV5, 6KUU	10.1038/s41564-019-0487-5
*Flaviviridae*	Classical swine fever virus (CSF)	5YF5, 5YF6, 5YF7, 5YF8, 6AE4, 6AE5, 6AE6, 6AE7	10.1093/nar/gky848
	Hepacivirus (HCV)	6MVO	10.1021/acs.jmedchem.8b01719
	Dengue virus 3 (DENV-3)	6XD0	10.1128/JVI.01130-20
	Zika virus (ZIKV)	6LD1 6UX2, 6WCZ	10.1128/JVI.00794-20 10.1038/s41594-020-0472-y
	Yellow fever virus 17D (YF)	6QSN	10.1016/j.antiviral.2019.104536
	Tick-borne encephalitis virus	7D6N	10.1093/nar/gkaa1250
*Picornaviridae*	Enterovirus D68 (EVD68)	6L4R	10.1016/j.jsb.2020.107510
	Enterovirus A71	6KWQ, 6KWR 6LSE, 6LSF, 6LSG, 6LSH	10.1093/nar/gkz1170 10.1038/s41467-020-16234-4
	Sicinivirus	6QWT	10.1016/j.jsb.2019.08.004
	Kobuvirus	6R1L	10.1016/j.jsb.2019.08.004
*Caliciviridae*	Murine norovirus (MNV)	5Y3D	10.3389/fmicb.2018.01466
*Coronaviridae*	Severe acute respiratory syndrome coronavirus 2 (SARS-CoV-2)	7CXN, 7CXM 7CYQ 7C2K, 7BZF 6XEZ 6YYT 7BW4 6NUR, 6NUS	10.1038/s41467-020-19770-1 10.1016/j.cell.2020.11.016 10.1016/j.cell.2020.05.034 10.1016/j.cell.2020.07.033 10.1038/s41586-020-2368-8 10.1016/j.celrep.2020.107774 10.1038/s41467-019-10280-3
*Permutotetraviridae*	Thosea asigna virus (TaV)	7OM2	10.3390/v13071260
*Reoviridae*	Bombyx mori cypovirus (BmCPV)	6TZ0, 6TZ1, 6TZ2, 6TY8, 6TY9	10.1038/s41594-019-0320-0
	Rotavirus A (RVA)	6OGY, 6OGZ	10.1038/s41467-019-10236-7
	Simian rotavirus A (SRV)	6OJ3, 6OJ4, 6OJ5, 6OJ6	10.1016/j.jmb.2019.06.016
	Bluetongue virus (BTV)	6PNS, 6PO2	10.1073/pnas.1905849116
	Cytoplasmic polyhedrosis virus (CPV)	6TY8, 6TY9, 6TZ0, 6TZ1, 6TZ2	10.1038/s41594-019-0320-0

**Table 2 viruses-14-02200-t002:** List of recently solved crystal and cryo-EM structures of RdRPs in complex with their respective allosteric inhibitors.

PDB ID	Virus	Ligand ID (PubChem)	Ligand Name	Site of Binding	Citation
6mvo	HCV	K4P	6-[(7-chloro-1-hydroxy-1\,3-dihydro-2\,1-benzoxaborol-5-yl)(methylsulfonyl)amino]-5-cyclopropyl-2-(4-fluorophenyl)-N-methyl-1-benzofuran-3-carboxamide	Palm	10.1021/ACS.JMEDCHEM.8B01719
6mvk	HCV	K4J	(4-{(4S)-3-[5-cyclopropyl-2-(4-fluorophenyl)-3-(methylcarbamoyl)-1-benzofuran-6-yl]-2-oxo-1\,3-oxazolidin-4-yl}-2-fluorophenyl)boronic acid	Palm	10.1021/ACS.JMEDCHEM.8B01719
6mvp	HCV	K4S	(4-{[5-cyclopropyl-2-(4-fluorophenyl)-3-(methylcarbamoyl)-1-benzofuran-6-yl](methylsulfonyl)amino}phenyl)boronic acid	Palm	10.1021/ACS.JMEDCHEM.8B01719
6mvq	HCV	K4M	(4-{1-[5-cyclopropyl-2-(4-fluorophenyl)-3-(methylcarbamoyl)-1-benzofuran-6-yl]-1H-1\,2\,4-triazol-5-yl}-2-fluorophenyl)boronic acid	Palm	10.1021/ACS.JMEDCHEM.8B01719
5qj0	HCV	J6D	6-[ethyl(methylsulfonyl)amino]-2-(4-fluorophenyl)-N-methyl-5-(3-{[1-(pyrimidin-2yl)cyclopropyl]carbamoyl}phenyl)-1-benzofuran-3-carboxamide	Palm	10.1021/ACSMEDCHEMLETT.8B00379
5qj1	HCV	J6J	6-(ethylamino)-2-(4-fluorophenyl)-5-(3-{[1-(5-fluoropyrimidin-2-yl)cyclopropyl]carbamoyl}-4-methoxyphenyl)-N-methyl-1-benzofuran-3-carboxamide	Palm	10.1021/ACSMEDCHEMLETT.8B00379
5zit	Enterovirus D68	NDP	nadph dihydro-nicotinamide-adenine-dinucleotide phosphate	Finger	10.1016/J.JSB.2020.107510
6xd1	Dengue virus 3	V0J	(2R)-4-(butyl{[2’-(1H-tetrazol-5-yl)[1\,1’-biphenyl]-4-yl]methyl}carbamoyl)-1-(2\,2-diphenylpropanoyl)piperazine-2-carboxylic acid	Between fingers and palm	10.1128/JVI.01130-20
6xd0	Dengue virus 3	V0M	2-[({2-[(2\,6-dichlorophenyl)amino]phenyl}acetyl)amino]-2\,3-dihydro-1h-indene-2-carboxylic acid	Between fingers and palm	10.1128/JVI.01130-20
6izx	Dengue virus 2	B5C	2-oxo-2H-1\,3-benzoxathiol-5-yl acetate	Palm	10.1371/JOURNAL.PNTD.0007894
6izz	Dengue virus 3	B5C	2-oxo-2H-1\,3-benzoxathiol-5-yl acetate	Palm	10.1371/JOURNAL.PNTD.0007894
6ld1	Zika virus	EDO	1\,2-Ethanediol	Between thumb and palm	10.1128/JVI.00794-20
6ld2	Zika virus	KY3	(1S\,2S\,4S\,5R)-2\,4-dimethoxy-5-thiophen-2-yl-cyclohexane-1-carboxylic acid	Between thumb and palm	10.1128/JVI.00794-20
6ld3	Zika virus	G8O,FB2	G8O : 2\,4-dimethoxy-5-thiophen-2-yl-benzoic acid, FB2: benzenesulfonamide	Between thumb and palm	10.1128/JVI.00794-20
6ld4	Zika virus	G8O,G8F	G8O : 2\,4-dimethoxy-5-thiophen-2-yl-benzoic acid,G8F : 3-methoxybenzenesulfonamide	Between thumb and palm	10.1128/JVI.00794-20
6ld5	Zika virus	G8O,G8L	G8O : 2\,4-dimethoxy-5-thiophen-2-yl-benzoic acid, G8L : quinoline-8-sulfonamide	Between thumb and palm	10.1128/JVI.00794-20
7d4f	SARS-CoV-2	H3U	8-(3-(3-aminobenzamido)-4-methylbenzamido)naphthalene-1\,3\,5-trisulfonic acid	Palm	[40]-X
7dfg	SARS-CoV-2	1RP	5-fluoro-2-oxo-1H-pyrazine-3-carboxamide	Palm	Unpublished
7aap7ctt	SARS-CoV-2	GE6	[[(2~{R},3~{S},4~{R},5~{R})-5-(3-aminocarbonyl-5-fluoranyl-2-oxidanylidene-pyrazin-1-yl)-3,4-bis(oxidanyl)oxolan-2-yl]methoxy-oxidanyl-phosphoryl] phosphono hydrogen phosphate	Palm	10.1073/PNAS.2021946118 10.1016/j.xinn.2021.100080
7dfh	SARS-CoV-2	RVP	[(2R,3S,4R,5R)-5-(3-carbamoyl-1,2,4-triazol-1-yl)-3,4-dihydroxyoxolan-2-yl]methyl dihydrogen phosphate	Palm	Unpublished
7bv1, 7bv2 7b3b, 7b3c, 7b3d 7dte 7c2k 7l1f	SARS-CoV-2	Remdesivir / F86	2-ethylbutyl (2S)-2-[[[(2R,3S,4R,5R)-5-(4-aminopyrrolo [2,1-f][1,2,4]triazin-7-yl)-5-cyano-3,4-dihydroxyoxolan-2-yl]methoxy-phenoxyphosphoryl]amino]propanoate / [(2~{R},3~{S},4~{R},5~{R})-5-(4-azanylpyrrolo [2,1-f][1,2,4]triazin-7-yl)-5-cyano-3,4-bis(oxidanyl)oxolan-2-yl]methyl dihydrogen phosphate	Palm	10.1126/science.abc1560 10.1038/s41467-020-20542-0 Unpublished 10.1016/j.cell.2020.05.034 10.1016/j.molcel.2021.01.035
7ozu	SARS-CoV-2	7OK	[(2~{R}\,3~{S}\,4~{R}\,5~{R})-5-[(4~{E})-4-hydroxyimino-2-oxidanylidene-pyrimidin-1-yl]-3\,4-bis(oxidanyl)oxolan-2-yl]methyl dihydrogen phosphate	Palm	10.1038/s41594-021-00651-0
7ozv	SARS-CoV-2	16B	[(2~{R}\,3~{S}\,4~{R}\,5~{R})-3\,4-bis(oxidanyl)-5-[4-(oxidanylamino)-2-oxidanylidene-pyrimidin-1-yl]oxolan-2-yl]methyl dihydrogen	Palm	10.1038/s41594-021-00651-0

**Table 3 viruses-14-02200-t003:** NADPH docked against different viral RdRPs in their apo and template-bound conformations.

RdRP	PDB ID	Interacting Residues (H-bond)	Binding Energy (kcal/mol)
Poliovirus RdRP apo	4r0e	Asp328, Leu175, Ser288, Ser294, Thr293, Gly292, Cys290, Arg188, Asp111, Lys127, and Ser184	−8.9
Poliovirus RdRP bound to RNA	4k4w	Glu108, Asp111, and template nucleotides	−10.6
EV71 RdRP apo	3n6l	Ser295, Gly290, Lys127, Ser184, Arg188, and Ser179	−8
EV71 RdRP bound to RNA	6kwq	Ala109, His113, and template nucleotides	−10.7
HCV RdRP apo	1nb4	Ser142, Lys141, Ala97, Gly449, Gly557, Gly558, and Asp559	−8.1
HCV RdRP bound to RNA	1nb7	Cys14, Lys141, Ser407, Glu398, and template nucleotides	−10
SARS-CoV-2 RdRP apo	7btf	Arg553, Arg555, Asp452, Arg624, Lys621, Pro620, Cys622, Tyr619, Asp760, Asp761, and Ala762	−7
SARS-CoV-2 RdRP bound to RNA	7c2k	Arg836, Lys500, Asn497, and template nucleotides	−10.4

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
