# Peer review of "Revisiting Viral RNA-Dependent RNA Polymerases: Insights from Recent Structural Studies"

_viruses, 2022, doi:10.3390/v14102200_

Round 1

Reviewer 1 Report

In this review, Kavitha Ramaswamy ,et al. present in comprehensively  many structures of RNA-dependent RNA polymerase (RdRP). RdRP represent a distinctive yet versatile class of nucleic acid polymerases encoded by RNA viruses for the replication and transcription of their genome.

The review aims at exploring these incongruities in the light of recent structural studies of RdRP in complex with diverse cofactors, RNA moieties, analogs and inhibitors.

Modification

1)    In the abstract, it is mentioned that the structure of the RdRP is comparable to that of a cupped right hand consisting of fingers, palm, and thumb subdomains. Schematic representation is needed here.

2)      IRdRPs (EC 2.7.7.48.) are ancient enzymes that catalyze the formation of phosphodiester bonds between ribonucleotides in an RNA template-dependent fashion. The chemical reaction should be detailed and a figure should be added.

Author Response

  1. In the abstract, it is mentioned that the structure of the RdRP is comparable to that of a cupped right hand consisting of fingers, palm, and thumb subdomains. Schematic representation is needed here.

As per your suggestion, we have now added a new figure (Figure 1b) marking all the structural details, including subdomains, motifs, and channels.

  1. RdRPs (EC 2.7.7.48.) are ancient enzymes that catalyze the formation of phosphodiester bonds between ribonucleotides in an RNA template-dependent fashion. The chemical reaction should be detailed and a figure should be added.

As per your suggestion, we have included the details of the reaction along with a figure. The details are included in the first paragraph of the section “Introduction” and read as: “The reaction involves the nucleophilic attack of the 3'-hydroxyl group of the nascent RNA on the α-phosphate of the incoming nucleoside triphosphate (NTP), resulting in the formation of a new phosphodiester bond and the release of a pyrophosphate moiety (Figure 1A).”

Reviewer 2 Report

The manuscript titled "Review of Viral RNA-Dependent RNA Polymerases: Insights from Recent Structural Studies" is a review of RNA-dependent RNA polymerases (RdRPs) that aims to highlight the different functions of RdRPs from different viruses.

From a strictly structural point of view, the authors briefly describe the RdRP core and then focus the review on the functions associated with the different N- and C-terminal protrusions or domains contained in each polymerase and their interaction with other proteins. RdRPs are an excellent target for antiviral development, then they add a section that summarizes the latest structures of RdRPs in complex with antivirals discussing each position. Finally, they propose NADPH as a broad-spectrum inhibitor after "in silico" analysis. In the conclusion, the authors highlight the importance of RdRP functions other than polymerization, RdRP inhibitors, and NADPH as a broad-spectrum antiviral.

The objective of the work seems to be correct and has great potential, but the manuscript has severe deficiencies that must be corrected. Authors should review the text, figures, and tables.

The text needs a new organization because it is difficult to read and unclear. To correct this, I believe that at least the following points need to be addressed:

In Introduction. The authors describe the core of RdRPs, but this description is too short. I recommend extending the introduction by explaining in more detail the different domains of the RdRPs, the conserved structural motifs and the functions in which they are involved. As well as the different entrance and exit tunnels of the RNAs, NTPs, etc…

Also, in line 32, the sentence 'the conserved aspartates of the motifs (D-x(4.5)-D and GDD..." is confusing. The conserved aspartates are located in motifs A and C.

The section “The business of copying” is divided into different subsections for each virus family. It would be clearer if it were grouped by domains or motifs that perform the same function, or even if it were grouped by N-terminal domains or C-terminal domains.

In addition, Table 1 contains errors that need to be corrected: murine norovirus is not a Picornaviridae, but a Calciviridae, SARS-CoV-2 is a Coronaviridae, and Thosea asigna virus is a Permutotetraviridae. And some structures are missing, like SARS-CoV-1 (6NUR of 2019) for example.

The figures are generally not clear. For example, Figure 1B uses the colors green and blue to superimpose two structures and it is impossible to observe any difference between the two very similar colors. Figure 1C shows an interaction between the VP3 protein and the bracelet domain of the RdRP, this interaction should be detailed. In Figure 4 it would be clearer if the conformational change was colored in a different color at each position while the rest of the structure could only be one of the two structures because they are identical. In some pictures, such as Figure 10, two structures are superimposed and one of them is shown as an opaque surface. The overlay would be clearer if the surface was semi-transparent. In conclusion, the figures should be clearer.

It is also important in each section to make a brief introduction to the new proteins or domains that will appear. For example, Section 2.1 talks about the importance of the bracelet domain, but doesn't introduce the bracelet domain, or highlights the importance of the interaction with the VP3 protein or the VP3A domain, I think a simple word or sentence that introduces the new protein would be helpful to the reader.

In the section “Picornaviral proteolytic paradigm”, the importance of N-terminal cleavage for polymerase activation has been previously described before the articles cited in this section, please check the bibliography.

In Inhibitors section. In table 2 enter citations where possible.

 In the Picornaviral Inhibitors section, line 445 I think the correct sentence is  at the interface between the fingers and the palm (not thumb). And the binding pocket described as unusual and novel was previously described with Coxsackie Virus B3 RdRP and the inhibitor GPC-N114 (PDB id 4Y34; van der Linden L, et al. (2015) The RNA template channel of the RNA-dependent RNA polymerase as a target for antiviral therapy development of multiple genera within one virus family (PLoS Pathog. 11(3):e1004733).

With all of the above, I think it is better for authors to have the opportunity to submit a manuscript in better conditions. I think it would be best for the authors to review the manuscript in depth and resubmit it when all these discrepancies (and more than likely the text presents) are addressed.

Author Response

  1. In Introduction. The authors describe the core of RdRPs, but this description is too short. I recommend extending the introduction by explaining in more detail the different domains of the RdRPs, the conserved structural motifs and the functions in which they are involved. As well as the different entrance and exit tunnels of the RNAs, NTPs, etc…

The necessary details of the different domains of the RdRPs, the conserved structural motifs and their functions, the different entrance and exit tunnels have been included in Section 1 Introduction in Page 1, Paragraph 2 and Page 2 Paragraph 1.

  1. Also, in line 32, the sentence 'the conserved aspartates of the motifs (D-x(4.5)-D and GDD..." is confusing. The conserved aspartates are located in motifs A and C.

The sentence is now removed and the details are included in paragraph one of Page 2 under Section 1 Introduction. It reads “Motif A houses the first aspartate of the catalytic motif DX2-4D.” and “The conserved GDD motif, which is used to bind metal ions, is located in the loop of motif C.”

  1. The section “The business of copying” would be clearer if it were grouped by domains or motifs that perform the same function, or even if it were grouped by N-terminal domains or C-terminal domains.

As per your suggestion, we have now reorganized the entire content under section 2, Business of replication, to keep the focus on the domains, motifs, and co-factors involved in the replication. The new subsections are:

2.1.Cooperative Co-factors

2.2.Amino terminal augmentations

2.3.C-terminal Bracelets

2.4.Motifs, pockets and more

  1. In addition, Table 1 contains errors that need to be corrected: murine norovirus is not a Picornaviridae, but a Calciviridae, SARS-CoV-2 is a Coronaviridae, and Thosea asigna virus is a Permutotetraviridae. And some structures are missing, like SARS-CoV-1 (6NUR of 2019) for example.

Thanks, the necessary editing has been done in Table 1 now and new PDBs (6XEZ

6YYT, 7BW4, 6NUR, and 6NUS have been added in Table 1 and PDBs corresponding to complexes of SARS-CoV-2 RdRP with Remidesivir, and Molnupiravir, are included in Table 2.

  1. The figures are generally not clear. For example, Figure 1B uses the colors green and blue to superimpose two structures and it is impossible to observe any difference between the two very similar colors. Figure 1C shows an interaction between the VP3 protein and the bracelet domain of the RdRP, this interaction should be detailed. In Figure 4 it would be clearer if the conformational change was colored in a different color at each position while the rest of the structure could only be one of the two structures because they are identical. In some pictures, such as Figure 10, two structures are superimposed and one of them is shown as an opaque surface. The overlay would be clearer if the surface was semi-transparent. In conclusion, the figures should be clearer.

We have reworked most of the figures. We hope they appear better, clearer, and more informative now.

  1. It is also important in each section to make a brief introduction to the new proteins or domains that will appear. For example, Section 2.1 talks about the importance of the bracelet domain, but doesn't introduce the bracelet domain, or highlights the importance of the interaction with the VP3 protein or the VP3A domain, I think a simple word or sentence that introduces the new protein would be helpful to the reader.

As per your suggestion, we have added a short description of the new proteins wherever applicable. For example, for the bracelet domain, the introduction reads “The λ3 polymerases of Reoviridae are enormous cage-like structures that are encircled by their long N- and C-terminal extensions. While the N-terminal region wraps the continuous surface between the fingers and thumb subdomains of the polymerase, the C-terminal domain is an annular structure, popularly referred to as the “bracelet domain”, with a large opening capable of admitting ds RNA.”

  1. In the section “Picornaviral proteolytic paradigm”, the importance of N-terminal cleavage for polymerase activation has been previously described before the articles cited in this section, please check the bibliography.

The relevant references are now included in the bibliography.

  1. doi:10.3389/FMICB.2019.01280/FULL.
  2. doi:10.1099/VIR.0.81802-0/CITE/REFWORKS.
  3. doi:10.1074/jbc.M707907200.
  4. doi:10.1128/JVI.35.2.414-419.1980.

  1. In Inhibitors section. In table 2 enter citations where possible.

The citations are now included.

  1. In the Picornaviral Inhibitors section, line 445 I think the correct sentence is at the interface between the fingers and the palm(not thumb). And the binding pocket described as unusual and novel was previously described with Coxsackie Virus B3 RdRP and the inhibitor GPC-N114 (PDB id 4Y34; van der Linden L, et al. (2015) The RNA template channel of the RNA-dependent RNA polymerase as a target for antiviral therapy development of multiple genera within one virus family (PLoS Pathog. 11(3):e1004733).

Thanks for pointing it out. We have corrected the text now and added the reference. It now reads as: “Previously, Picornaviridae inhibitor complexes were shown to target antivirals at the interface of the fingers and the palm subdomains [4]. The EVD68 RdRP crystal structures in the apo state (PDB: 6L4R) and in complex with the non-nucleoside-like inhibitor NADPH (PDB: 5ZIT, Ligand ID NDP) revealed a binding pocket at the fingers subdomain close to the RNA template channel that correlated to a similar one in the Coxsackie Virus B3 RdRP [84,85].”

Round 2

Reviewer 2 Report

The authors have corrected the errors previously mentioned and have improved both the text and most of the figures.